# Aligning Transformers with Continuous Feedback via Energy Rank Alignment

**Shriram Chennakesavalu**
Department of Chemistry
Stanford University
Stanford CA, 94305
shriramc@stanford.edu

**Frank Hu**
Department of Chemistry
Stanford University
Stanford CA, 94305
frankhu@stanford.edu

**Sebastian Ibarraran**
Department of Chemistry
Stanford University
Stanford CA, 94305
sebastian.ibarraran@stanford.edu

**Grant M. Rotskoff**
Department of Chemistry
Stanford University
Stanford CA, 94305
rotskoff@stanford.edu

## Abstract

Searching through chemical space is an exceptionally challenging problem because the number of possible molecules grows combinatorially with the number of atoms. Large, autoregressive models trained on databases of chemical compounds have yielded powerful generators, but we still lack robust strategies for generating molecules with desired properties. This molecular search problem closely resembles the "alignment" problem for large language models, though for many chemical tasks we have a specific and easily evaluable reward function. Here, we introduce an algorithm called energy rank alignment (ERA) that leverages an explicit reward function to produce a gradient-based objective that we use to optimize autoregressive policies. We show theoretically that this algorithm is closely related to proximal policy optimization (PPO) and direct preference optimization (DPO), but has a minimizer that converges to an ideal Gibbs-Boltzmann distribution with the reward playing the role of an energy function. Furthermore, this algorithm is highly scalable, does not require reinforcement learning, and performs well relative to DPO when the number of preference observations per pairing is small. We deploy this approach to align molecular transformers and protein language models to generate molecules and protein sequences, respectively, with externally specified properties and find that it does so robustly, searching through diverse parts of chemical space.

## 1 Introduction

Large language models (LLMs) are trained on large corpora of text to autoregressively generate outputs. These models strongly reflect the distribution of the data on which they are trained Ouyang et al. [2022], and controlling the outputs to reflect externally imposed preferences is an increasingly important challenge for deployment. The aforementioned task, often called "alignment", requires either careful curation of training data or large sets of human preference data—both options are labor-intensive Casper et al. [2023]. Reinforcement learning from human feedback (RLHF), a family of algorithms that employs these human preference datasets, has been widely employed to align instruction and chat models Ouyang et al. [2022], Bai et al. [2022], but it is both expensive to

acquire the training data and difficult to carry out in practice Casper et al. [2023]. Recent algorithmic developments, such as direct preference optimization (DPO) Rafailov et al. [2023], simplify the alignment framework by making the reward function implicit, but still require human preference data. While these algorithms succeed in constraining outputs, many "alignment"-like tasks require evaluation that would be difficult for human evaluators.

Generative sampling problems seeking to optimize a reward are common in chemistry, where comparing small molecules using a particular functional assay or computationally accessible property is often far easier than searching chemical space to identify novel compounds. Recent efforts to build large, domain-specific models for chemistry Chithrananda et al. [2020] have shown promising performance on both property prediction and reaction prediction tasks. Nevertheless, just as with LLMs, leveraging these models for molecule optimization requires first guiding "unaligned" models to favor important properties like synthetic accessibility or solubility. Here, we seek to productively search chemical space using transformers by introducing a new preference optimization algorithm, which we call energy rank alignment.

**Our contribution:** We formulate a generic alignment algorithm that we call *Energy Rank Alignment* or ERA that leverages an explicit reward function to guide autoregressive sampling while targeting specific properties or preferences. Unlike reward maximization in RL-based algorithms, the policy that minimizes our objective is designed to sample fluctuations around a maximal reward value to promote sample diversity. Our algorithm enables direct gradient-based optimization of a policy to match the ideal preference distribution and converges asymptotically to an optimal distribution with tuneable entropy and controllable regularization, which we show theoretically. The minimizers of our objective are closely related to the minimizer of PPO and DPO, but we have more direct control over the influence of the regularization relative to fluctuations around the maximum reward. In numerical experiments, we demonstrate that this algorithm successfully aligns molecular transformer model to identify a highly diverse set of chemicals with properties favored by our choice of reward. Finally, we demonstrate that ERA is able to align a protein language model to generate mutated protein sequences with desirable properties according to a computational reward model.

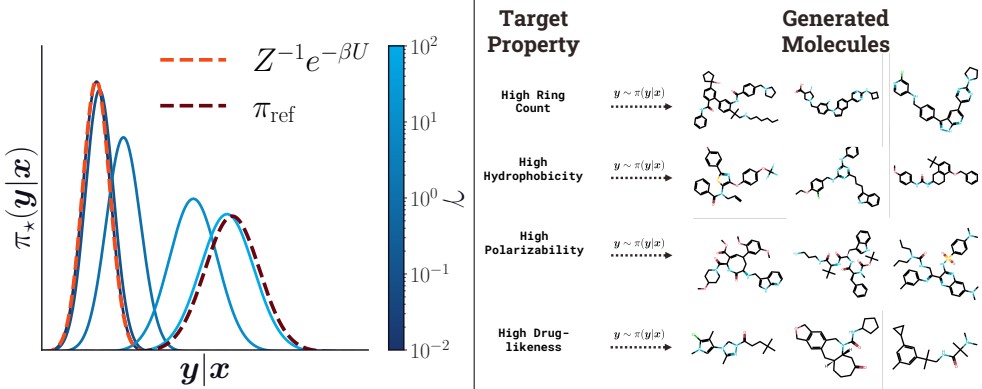

Figure 1: Energy rank alignment (ERA) enables targeting low-energy, high-reward regions with controllable fluctuations. Optimal policy approaches Boltzmann distribution with low regularization ($\gamma \to 0$) and reference policy with high regularization ($\gamma \to \infty$) (left). Aligned models can be used to sample molecules with desired chemical properties (right).

## 1.1 Related Work

Inverse molecular design tasks have a long history Lindsay et al. [1993] and many recent works have sought to apply machine learning to facilitate this difficult search problem Sanchez-Lengeling and Aspuru-Guzik [2018], Gromski et al. [2019], Gómez-Bombarelli et al. [2018]. While reinforcement learning has proved a popular strategy for molecular optimization Zhou et al. [2023], Sanchez-Lengeling and Aspuru-Guzik [2018], several recent studies have sought to use transformers Vaswani et al. [2017] trained on large databases of molecules represented with the text-based SMILES syntax Chithrananda et al. [2020], Schwaller et al. [2018], Wang et al. [2019], Bagal et al. [2022] for such tasks. Schwaller et al. [2019] utilized an atom-wise tokenization, which we also employ, to

train a transformer for the downstream task of reaction prediction. These "chemical language models" have been studied for applications on downstream tasks, including property prediction Bagal et al. [2022], Chithrananda et al. [2020] and reaction prediction Pesciullesi et al. [2020], Schwaller et al. [2018].

Building scalable strategies for alignment has attracted enormous attention because of the high cost and complexity of constraining LLM outputs. Much of the current paradigm is built on reinforcement learning from human feedback (RLHF) Ouyang et al. [2022]. Within this framework, human preferences provided in the form of pairwise rankings are first used to train a reward model, and subsequently that reward model is used to optimize a policy using, for example, proximal policy optimization (PPO) Schulman et al. [2017]. Rafailov et al. [2023] demonstrated that the reward model can be treated implicitly using a scheme that maximizes the likelihood of the preferences given an offline dataset. Because this approach does not require training a reward model, it has been named Direct Preference Optimization (DPO). Our work differs from both strategies; first, unlike RLHF, we do not employ reinforcement learning and instead develop an explicit, gradient-based objective for the optimal policy. Secondly, unlike DPO, we leverage an explicit reward function and add regularization transparently, both of which help to avoid greedy policies Azar et al. [2023]. However, like both approaches, we assume that the Bradley-Terry model Bradley and Terry [1952] of preference data is appropriate for the underlying target distribution.

Many recent works have built upon the ideas of RLHF and DPO, including studies on the effect of point-wise sampling of preference distributions Azar et al. [2023], investigations into the theoretical basis for contrastive methods for unlearning target datasets Zhang et al. [2024], and alternatives to the Bradley-Terry pairwise preference model Munos et al. [2023], An et al. [2023]. One recent study explores alignment in the context of inverse molecular design: Park et al. [2023] applies DPO to SMILES generators to increase the probability of activity for generated compounds against a drug target. However, they indicate that many preferences in chemistry are expressed as continuous signals, which is not suitable for DPO. Overcoming this limitation while maintaining the advantages of a direct gradient-based policy optimization strategy is a central goal of our current work. Our analysis and methodology directly addresses issues related to point-wise sampling because the explicit reward function eliminates overly greedy assignments of preference probabilities. Indeed, as discussed in Sec. 4, we see that DPO mode collapses where ERA shifts the policy towards the target distribution. While non-transitive preferences may arise in some settings, leading to a breakdown of the Bradley-Terry preference distribution model, by construction our target rewards are determined by quantitative evaluations of properties, and are therefore transitive.

## 2 Energy rank alignment

A policy is a conditional probability distribution $\pi(\cdot|\boldsymbol{x}) : \mathcal{Y} \to \mathbb{R}$; we generate an output $\boldsymbol{y}$ from prompt $\boldsymbol{x}$. The spaces $\mathcal{Y}$ and $\mathcal{X}$ are discrete and finite, corresponding to sequences of tokenized outputs of the model with a maximum length. In alignment tasks, we begin with a pre-trained reference policy $\pi_{\mathrm{ref}}$ and seek to optimize a parametric, trainable policy $\pi_{\boldsymbol{\theta}}$ to adapt the conditional sampling for a particular task or constraint.

Consider a prompt $\boldsymbol{x} \in \mathcal{X}$ and model outputs $\boldsymbol{y}, \boldsymbol{y}' \in \mathcal{Y}$ and a collection of preferences $\mathcal{D} = \{(\boldsymbol{y}_i \succ \boldsymbol{y}_i'; \boldsymbol{x}_i)\}_{i=1}^n$; the notation $\succ$ indicates that $\boldsymbol{y}_i$ is preferred to $\boldsymbol{y}_i'$. The conditional probability that $\boldsymbol{y} \succ \boldsymbol{y}'$ given $\boldsymbol{x}$ can be modeled as a pairwise Boltzmann ranking within the Bradley-Terry model, i.e.,

$$p(\boldsymbol{y} \succ \boldsymbol{y}'|\boldsymbol{x}) = \frac{e^{-\beta U(\boldsymbol{x},\boldsymbol{y})}}{e^{-\beta U(\boldsymbol{x},\boldsymbol{y})} + e^{-\beta U(\boldsymbol{x},\boldsymbol{y}')}} \equiv \sigma\big(\beta U(\boldsymbol{x},\boldsymbol{y}') - \beta U(\boldsymbol{x},\boldsymbol{y})\big). \tag{1}$$

Here $\beta > 0$ is a constant, $\sigma(x) = (1 + e^{-x})^{-1}$ and we refer to $U : \mathcal{X} \times \mathcal{Y} \to \mathbb{R}$ as an energy function to make clear the connection to statistical physics, but it is the negative reward within the RL framework for alignment.

To impose the preferences we minimize the objective

$$J(\pi) = \mathbb{E}_{\boldsymbol{x} \sim \nu}\left[\int U(\boldsymbol{x},\boldsymbol{y})\mathrm{d}\pi(\boldsymbol{y}|\boldsymbol{x}) + \beta^{-1}\int (1+\gamma)\log \pi(\boldsymbol{y}|\boldsymbol{x}) - \gamma\log(\pi_{\mathrm{ref}}(\boldsymbol{y}|\boldsymbol{x}))\mathrm{d}\pi(\boldsymbol{y}|\boldsymbol{x})\right], \tag{2}$$

where $\beta^{-1}$ is a parameter controlling the magnitude of the entropic term, $\gamma$ sets the scale of the Kullback-Leibler regularization compared with the energy term, and $\nu$ is a probability distribution

over the prompts $\nu \in \mathcal{P}(\mathcal{X})$. A proximal scheme for gradient descent on this objective corresponds to a gradient flow on $J$ Santambrogio [2017], Maas [2011]; the functional can be viewed as a free energy, and the corresponding flow is

$$\partial_t \pi_t = \nabla \cdot (\pi_t \nabla \delta_\pi J[\pi_t]), \tag{3}$$

and $\delta_\pi$ denotes the Fréchet derivative with respect to $\pi$. Assuming that $\pi_0$ has full support on $\mathcal{X} \times \mathcal{Y}$, the optimization converges asymptotically to a stationary policy which satisfies

$$\nabla \delta_\pi J[\pi_\star] = 0 \iff \pi_\star \propto e^{-\frac{\beta}{1+\gamma} U + \frac{\gamma}{\gamma+1} \log \pi_{\text{ref}}}, \tag{4}$$

and this minimizer is globally optimal. In the context of LLM alignment, a representation of the energy function $U : \mathcal{X} \times \mathcal{Y} \to \mathbb{R}$ is learned as a "reward model", though we also consider tasks in which $U$ is an easily evaluated function of the pair $(\boldsymbol{x}, \boldsymbol{y})$. The optimal distribution $\pi_\star$ is a Gibbs-Boltzmann measure

$$\pi_\star(\boldsymbol{y}|\boldsymbol{x}) = Z^{-1}(\boldsymbol{x}) \exp\left[-\frac{\beta}{1+\gamma}\big(U(\boldsymbol{x}, \boldsymbol{y}) - \beta^{-1}\gamma \log \pi_{\text{ref}}(\boldsymbol{y}|\boldsymbol{x})\big)\right] \tag{5}$$

where $Z(\boldsymbol{x})$ is the $\boldsymbol{x}$-dependent normalization constant. This expression makes clear the effect of $\beta$: when $\beta \to \infty$ (low temperature), the reward dominates and fluctuations around the maximal reward are small, which could lead to "mode-seeking"; when $\beta \to 0$ (high physical temperature) fluctuations around the maximal reward increase and the regularization term favors proximity to $\pi_{\text{ref}}$. Similarly, $\gamma \to 0$ recovers a Gibbs-Boltzmann distribution proportional to $e^{-\beta U}$ at inverse temperature $\beta$, while $\gamma \to \infty$ is dominated by the reference policy.

**Loss functions for $\pi_\theta$:** Proximal Policy Optimization (PPO) optimizes an indirect, proximal objective to minimize an objective closely related to (2) (cf. Appendix 3, A). Direct Preference Optimization (DPO) treats the negative reward function $U$ implicitly and directly maximizes the likelihood of $p(\boldsymbol{y} \succ \boldsymbol{y}'|\boldsymbol{x})$. Our objectives differ from both approaches: like DPO, we directly optimize the policy using an explicit, gradient-based objective, but, in contrast, we use a reward function directly in our objective. The losses we build are thus amenable to both offline (samples from $\pi_{\text{ref}}$) and online (samples from $\pi_\theta$) policy alignment, as explained below. Choosing to optimize the objective online has been shown to have important consequences on performance Tajwar et al. [2024], though we focus here on the setting where samples are drawn offline.

We directly optimize the Kullback-Leibler divergence between the entropy-regularized preference distribution $p_\gamma(\boldsymbol{y} \succ \boldsymbol{y}'|\boldsymbol{x})$ and the corresponding parametric preference distribution $p_\theta(\boldsymbol{y} \succ \boldsymbol{y}'|\boldsymbol{x})$. Explicitly, using the fact that conditional preference distribution is normalized, we obtain

$$\begin{aligned} D_{\text{KL}}^{(\boldsymbol{y},\boldsymbol{y}')}(p_\gamma|p_\theta) &= p_\gamma(\boldsymbol{y} \succ \boldsymbol{y}'|\boldsymbol{x}) \log \frac{p_\gamma(\boldsymbol{y} \succ \boldsymbol{y}'|\boldsymbol{x})}{p_\theta(\boldsymbol{y} \succ \boldsymbol{y}'|\boldsymbol{x})} + p_\gamma(\boldsymbol{y}' \succ \boldsymbol{y}|\boldsymbol{x}) \log \frac{p_\gamma(\boldsymbol{y}' \succ \boldsymbol{y}|\boldsymbol{x})}{p_\theta(\boldsymbol{y}' \succ \boldsymbol{y}|\boldsymbol{x})}, \\ &= p_\gamma(\boldsymbol{y} \succ \boldsymbol{y}'|\boldsymbol{x}) \log \frac{p_\gamma(\boldsymbol{y} \succ \boldsymbol{y}'|\boldsymbol{x})}{p_\theta(\boldsymbol{y} \succ \boldsymbol{y}'|\boldsymbol{x})} + \big(1 - p_\gamma(\boldsymbol{y} \succ \boldsymbol{y}'|\boldsymbol{x})\big) \log \frac{1 - p_\gamma(\boldsymbol{y} \succ \boldsymbol{y}'|\boldsymbol{x})}{1 - p_\theta(\boldsymbol{y} \succ \boldsymbol{y}'|\boldsymbol{x})}, \end{aligned} \tag{6}$$

where

$$p_\gamma := \sigma\left(\frac{\beta}{1+\gamma}\left[(U(\boldsymbol{x}, \boldsymbol{y}') - U(\boldsymbol{x}, \boldsymbol{y})) + \beta^{-1}\gamma \log \frac{\pi_{\text{ref}}(\boldsymbol{y}|\boldsymbol{x})}{\pi_{\text{ref}}(\boldsymbol{y}'|\boldsymbol{x})}\right]\right). \tag{7}$$

This quantity is a well-defined KL divergence and is hence non-negative; the quantity vanishes when $p_\gamma = p_\theta$ on the observations $\boldsymbol{y}, \boldsymbol{y}'$. Furthermore, with access to an explicit reward model, all terms in (6) can be computed directly and

$$p_\theta(\boldsymbol{y} \succ \boldsymbol{y}'|\boldsymbol{x}') = \frac{\pi_\theta(\boldsymbol{y}|\boldsymbol{x})}{\pi_\theta(\boldsymbol{y}|\boldsymbol{x}) + \pi_\theta(\boldsymbol{y}'|\boldsymbol{x})} = \sigma\left(\log \frac{\pi_\theta(\boldsymbol{y}|\boldsymbol{x})}{\pi_\theta(\boldsymbol{y}'|\boldsymbol{x})}\right). \tag{8}$$

To obtain a minimizer of the regularized objective defined in (2) we optimize

$$\mathcal{L}^{\text{ERA}}(\pi_\theta) = \mathbb{E}_{x \sim \mathcal{D}} \mathbb{E}_{\boldsymbol{y}, \boldsymbol{y}' \sim \pi_{\text{ref}}(\cdot|\boldsymbol{x})} D_{\text{KL}}^{(\boldsymbol{y},\boldsymbol{y}')}(p_\gamma|p_\theta); \tag{9}$$

If the current policy overlaps with the target preference distribution, it may be useful to sample directly from the partially aligned policy, i.e., to use the "on-policy" formulation,

$$\mathcal{L}_{\text{on}}^{\text{ERA}}(\pi_\theta) = \mathbb{E}_{\boldsymbol{x} \sim \mathcal{D}} \mathbb{E}_{\boldsymbol{y}, \boldsymbol{y}' \sim \pi_\theta(\boldsymbol{y}|\boldsymbol{x})} D_{\text{KL}}^{(\boldsymbol{y},\boldsymbol{y}')}(p_\gamma|p_\theta) \tag{10}$$

instead of (9). One issue that arises with this scheme is differentiation with respect to the parameters of the policy $\boldsymbol{\theta}$ because $\boldsymbol{y}$ and $\boldsymbol{y}'$ are decoded into discrete tokens, an operation that is not differentiable. To remedy this, we importance sample with a reference policy

$$\mathcal{L}_{\text{on}}^{\text{ERA}}(\pi_{\boldsymbol{\theta}}) = \mathbb{E}_{\boldsymbol{x} \sim \mathcal{D}} \mathbb{E}_{\boldsymbol{y}, \boldsymbol{y}' \sim \pi_{\text{ref}}(\boldsymbol{y}|\boldsymbol{x})} \frac{\pi_{\boldsymbol{\theta}}(\boldsymbol{y}|\boldsymbol{x})\pi_{\boldsymbol{\theta}}(\boldsymbol{y}'|\boldsymbol{x})}{\pi_{\text{ref}}(\boldsymbol{y}|\boldsymbol{x})\pi_{\text{ref}}(\boldsymbol{y}'|\boldsymbol{x})} D_{\text{KL}}^{(\boldsymbol{y}, \boldsymbol{y}')}(p_{\gamma}|p_{\boldsymbol{\theta}}). \tag{11}$$

This reweighting is straightforward and the importance weights should generally be appreciable, especially early in training when $\pi_{\boldsymbol{\theta}}$ has not drifted far from $\pi_{\text{ref}}$. It is, of course, also natural to iteratively update $\pi_{\boldsymbol{\theta}}$ using a previous iterate as the reference policy. In this work, we only use (9) as an objective and leave the on-policy objectives to future work. For an ablation of the parameters of ERA and a direct comparison in task performace to DPO, see Section C.3.1.

## 3 Theoretical Analysis

To understand the ERA loss function and its connection to the entropy regularized objective (2), we begin by establishing that the minimizers of (9) are of the form (5). We first define the notion of equivalence precisely.

**Definition 3.1** *The conditional probability measures $\pi(\cdot|\boldsymbol{x})$ and $\pi'(\cdot|\boldsymbol{x})$ are conditionally equivalent if $\forall \boldsymbol{x} \in \mathcal{X}$, $\pi$ and $\pi'$ are such that $\sup_{\boldsymbol{y} \in \mathcal{Y}} |\pi(\boldsymbol{y}|\boldsymbol{x}) - \pi'(\boldsymbol{y}|\boldsymbol{x})| = 0$.*

We remark that this strong form of equivalence is appropriate on the finite, discrete spaces $\mathcal{X}$ and $\mathcal{Y}$ we consider here.

**Lemma 3.1** *If $\pi$ is conditionally equivalent to $\pi'$, then $\pi'_g(\cdot|\boldsymbol{x}) \propto \pi'(\cdot|\boldsymbol{x})e^{g(\boldsymbol{x})}$ is conditionally equivalent to $\pi$ for all functions $g : \mathcal{X} \to \mathbb{R}$ such that $\sup_{\boldsymbol{x} \in \mathcal{X}} |e^{g(\boldsymbol{x})}| < +\infty$.*

We prove Lemma 3.1 in Appendix A and use this simple lemma to prove the following result.

**Proposition 3.2** *Suppose $\pi(\cdot|\boldsymbol{x}) \in \mathcal{P}(\mathcal{Y})$ and that $\text{supp}(\pi) = \text{supp}(\pi_{\text{ref}})$. Let $\beta > 0$, $\gamma \geq 0$ and that the reward model is such that $\sup_{\boldsymbol{x}, \boldsymbol{y} \in \mathcal{X} \times \mathcal{Y}} |e^{-U(\boldsymbol{x}, \boldsymbol{y})}| < +\infty$. Then, the minimizer of $\mathcal{L}^{\text{ERA}}$ is conditionally equivalent to $\pi_{\star}$.*

First, we verify that any probability measure $\pi_g(\boldsymbol{y}|\boldsymbol{x}) \propto \exp(-\frac{\beta}{1+\gamma}(U(\boldsymbol{x}, \boldsymbol{y}) - \beta^{-1}\gamma \log \pi_{\text{ref}}(\boldsymbol{y}|\boldsymbol{x})) + g(\boldsymbol{x}))$ minimizes the objective. Because $\mathcal{L}^{\text{ERA}}$ is non-negative, it suffices to show that for all pairs $\boldsymbol{y}, \boldsymbol{y}'$, $D_{\text{KL}}^{(\boldsymbol{y}, \boldsymbol{y}')}(p_{\gamma}|p_{\boldsymbol{\theta}}) \equiv 0$. This follows immediately from the cancellation in the preference probability $p_{\gamma}$ of $e^{g(\boldsymbol{x})}$ after factorization in (5).

Now, suppose that $\pi(\boldsymbol{y}|\boldsymbol{x}) \neq \exp\left(-\frac{\beta}{1+\gamma}(U(\boldsymbol{x}, \boldsymbol{y}) - \beta^{-1}\gamma \log \pi_{\text{ref}}(\boldsymbol{y}|\boldsymbol{x}))\right)$ where we have taken $g(\boldsymbol{x}) = 0$ without loss of generality and $\pi := \pi_g$. Assume that for all pairs $\boldsymbol{y}, \boldsymbol{y}'$, the divergence $D_{\text{KL}}^{(\boldsymbol{y}, \boldsymbol{y}')}(p_{\gamma}|p_{\boldsymbol{\theta}}) \equiv 0$ which is required of a minimizer. Equivalently, it must be the case that for all $\boldsymbol{y}, \boldsymbol{y}'$,

$$\frac{\pi(\boldsymbol{y}|\boldsymbol{x})}{\pi(\boldsymbol{y}|\boldsymbol{x}) + \pi(\boldsymbol{y}'|\boldsymbol{x})} = \frac{\pi_{\star}(\boldsymbol{y}|\boldsymbol{x})}{\pi_{\star}(\boldsymbol{y}|\boldsymbol{x}) + \pi_{\star}(\boldsymbol{y}'|\boldsymbol{x})} \implies \frac{\pi(\boldsymbol{y}'|\boldsymbol{x})}{\pi(\boldsymbol{y}|\boldsymbol{x})} = \frac{\pi_{\star}(\boldsymbol{y}'|\boldsymbol{x})}{\pi_{\star}(\boldsymbol{y}|\boldsymbol{x})}, \tag{12}$$

from which we see that

$$\pi(\boldsymbol{y}|\boldsymbol{x}) = \frac{\pi(\boldsymbol{y}'|\boldsymbol{x})}{e^{-\frac{\beta}{1+\gamma}(U(\boldsymbol{x}, \boldsymbol{y}') - \beta^{-1}\gamma \log \pi_{\text{ref}}(\boldsymbol{y}'|\boldsymbol{x}))}} e^{-\frac{\beta}{1+\gamma}(U(\boldsymbol{x}, \boldsymbol{y}) - \beta^{-1}\gamma \log \pi_{\text{ref}}(\boldsymbol{y}|\boldsymbol{x}))}. \tag{13}$$

By construction, $\pi(\boldsymbol{y}|\boldsymbol{x})$ does not depend on $\boldsymbol{y}'$ so the prefactor must be purely a function of $\boldsymbol{x}$, which completes the proof, using Lemma 3.1. A detailed theoretical analysis of the ERA objective is provided in Appendix A. This analysis compares the ERA loss with the DPO and PPO objectives and demonstrates that ERA yields correct global rankings in the low data limit, while DPO maximizes pairwise margin.

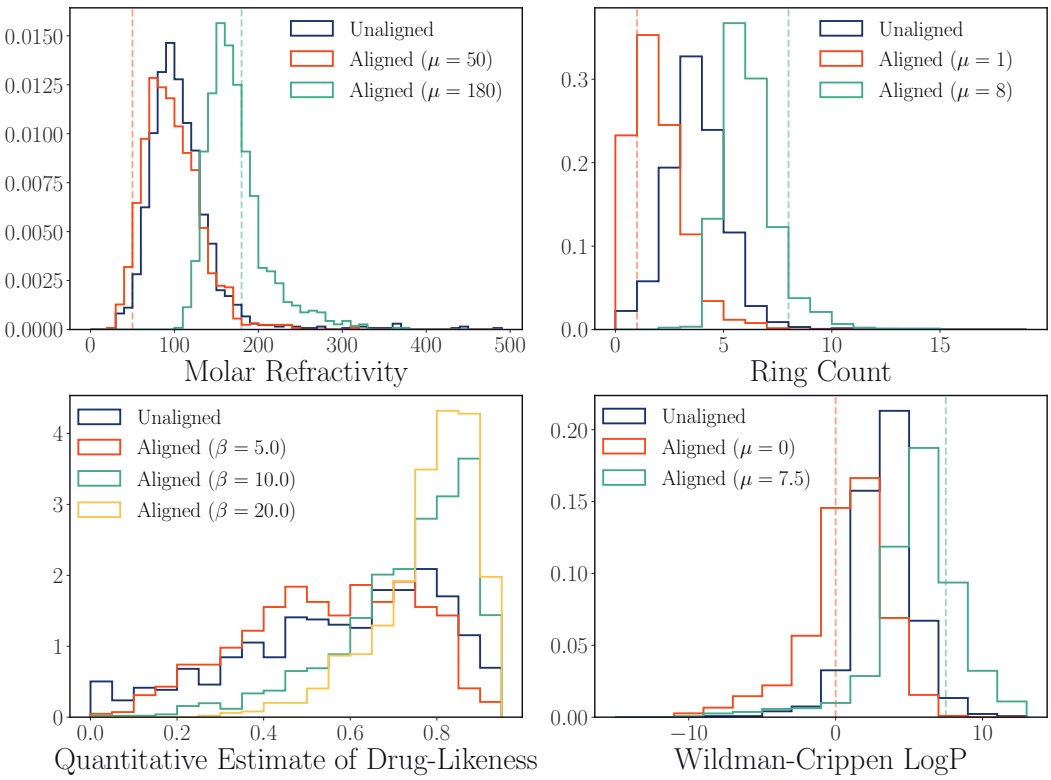

Figure 2: Unprompted molecular generator alignment. Distributions of different chemical properties for molecules sampled from aligned and unaligned policies. The center of the harmonic potential, $\mu$, is varied for MR ($\beta = 1.0$), Ring Count ($\beta = 1.0$), and LogP ($\beta = 10.0$), while $\beta$ is varied for QED. All experiments were run with no regularization to the reference policy ($\gamma = 0$).

## 4 Experiments

We test ERA on both chemical and language tasks to shed light on the following questions: 1) Can we use ERA to robustly fine-tune our model to generate samples according to a desired distribution? 2) What is the effect of changing the inverse-temperature $\beta$ during ERA? 3) Do we maintain sample diversity (and validity) without regularizing to remain close to a reference policy, and what is the effect of increased regularization? 4) Can we simultaneously target multiple properties with high fidelity, and how can we trade off between desired properties?

### 4.1 Generating small molecules with desired properties

We use a decoder-only representation for the molecular generator Bagal et al. [2022], where the generator has 2 layers, an embedding dimension of 512, a vocabulary of 324 tokens, and totals 3.5M parameters. Starting from a random initialization, we carry out pretraining on a dataset of 2.4M small molecules from the ChEMBL database Zdrazil et al. [2024] for 180 epochs. For sampling from our molecular generator, we use top-$k$ sampling with $k = 5$ and a sampling temperature of $T = 1$ in all experiments for consistency. This version of the model is not conditioned on a prompt and generates a small molecule (represented as a SMILES string) given just a start-of-sequence token.

Central to ERA is, of course, access to a computable energy function. As a proof-of-concept, we first consider 5 different properties for which the corresponding energy function is easily evaluable: Quantitative Estimate of Drug-Likeness (QED) Bickerton et al. [2012], Wildman-Crippen LogP (LogP) Wildman and Crippen [1999], Ring Count, Molar Refractivity (MR) Wildman and Crippen [1999], and Tanimoto Similarity Rogers and Tanimoto [1960] (Section 4.1.1, 4.1.2). Briefly, LogP is a measure of the hydrophobicity of a molecule, MR is a measure of the polarizability of the molecule, and Tanimoto similarity is a measure of the similarity between two molecules (see Appendix C.2). We

then investigate ERA in a more challenging context: generating small-molecules with high predicted bioactivity for two kinases (Section 4.1.3).

### 4.1.1 Unprompted molecular alignment on RDKit oracles

First, we independently target four different properties using ERA with an unprompted molecular generator (Fig. 2). Using the pretrained model as our reference policy, we generate a dataset $\mathcal{D} = \{\boldsymbol{y}_1^{(i)}, \boldsymbol{y}_2^{(i)}, U(\boldsymbol{y}_1^{(i)}), U(\boldsymbol{y}_2^{(i)})\}_{i=1}^N$ and carry out energy rank alignment on $\pi_{\boldsymbol{\theta}}$, where $\pi_{\boldsymbol{\theta}}$ is initialized using the weights of $\pi_{\text{ref}}$. Here, $\boldsymbol{y}_1, \boldsymbol{y}_2 \sim \pi_{\text{ref}}$ and $\boldsymbol{y}$ and $U(\boldsymbol{y})$ denote the generated molecule and its corresponding energy, respectively. For MR, Ring Count, and LogP, we define the energy $U$ to be a harmonic potential centered at a target value. For QED, we define the energy to be the negative logarithm of QED and vary $\beta$ to assess its impact on alignment (see Tables S2, S3). In Fig. 2, we see that we successfully shift the distribution to target means that are both greater and lower than the average value of MR, Ring Count, and LogP under the reference policy. Furthermore, in the alignment of QED, we observe the effect of changing $\beta$ on the learned policy; with increased $\beta$, the learned policy concentrates around low-energy samples (i.e. near QED = 1), and with lower $\beta$, the learned policy samples a greater range of QED values, as expected. We note that for each of these four experiments, we did not regularize towards the reference policy (i.e. $\gamma = 0$). Even so, we were able to maintain both sample diversity and appreciable sample validity (see Fig. S8 and Table S6).

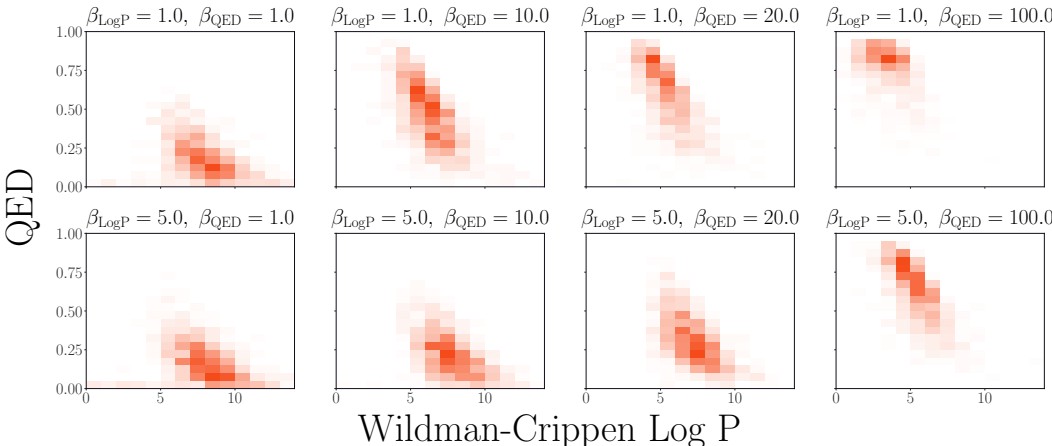

Figure 3: Unprompted multi-property molecular generator alignment. 2D histograms of LogP versus QED for different combinations of property-specific $\beta$ illustrating a clear trade-off when performing multi-property alignment. Relative increases in $\beta$ for a given property target higher values for that property. All experiments were run with no regularization to the reference policy ($\gamma = 0$).

Many molecular design tasks require balancing multiple properties, and designing an objective for multi-property alignment is straightforward within the ERA framework. To demonstrate this, we generate molecules with both high QED and LogP using ERA with an energy function weighted by property-specific $\beta$: $U = \beta_{\text{QED}} U_{\text{QED}} + \beta_{\text{LogP}} U_{\text{LogP}}$ (see Tables S2, S7 for details on the energy function). We carry out ERA with different pairs of $(\beta_{\text{QED}}, \beta_{\text{LogP}})$ using the same procedure as above, and from Fig. 3, we see that we target multiple properties with varying fidelity by simply modulating the value of property-specific $\beta$. Ultimately, increasing the $\beta$ for an individual property enables us to favor higher values of that property in multi-property alignment setting. In this case, we also do not regularize with the KL-divergence to the reference policy and again maintain sample diversity and validity (see Fig. S9 and Table S7).

### 4.1.2 Prompted molecular alignment on RDKit oracles

Inspired by the task of lead optimization in drug discovery efforts Keserü and Makara [2009], we ask whether we can use ERA to train a molecular generator that can sample a molecule that is both similar to the prompt molecule *and* also exhibits some desired property. First, we fine-tune the pretrained molecular generator to enable prompted molecular generation (see Appendix C.3.3) and use this fine-tuned model as our reference policy for all prompted molecular alignment tasks. This reference

policy disproportionately samples molecules that are identical (i.e. a Tanimoto similarity of 1.0) to the prompt molecule (see Fig. S10), so we carry out multi-property alignment on this reference policy to generate molecules that are similar—but not identical—to the prompt molecule and also have a high drug-likeness as measured by QED. Using ERA, we optimize the reference policy with a generated dataset $\mathcal{D} = \{(\boldsymbol{y}_1^{(i)}, \boldsymbol{x}^{(i)}), (\boldsymbol{y}_2^{(i)}, \boldsymbol{x}^{(i)}), U(\boldsymbol{y}_1^{(i)}, \boldsymbol{x}^{(i)}), U(\boldsymbol{y}_2^{(i)}, \boldsymbol{x}^{(i)})\}_{i=1}^N$, where we sample four molecules for each prompt molecule from the reference policy and consider all possible preference pairs for a total of six preference pairs per prompt molecule (see Appendix C.2 for full details on the energy used).

We observe that the per-prompt average QED under the optimized policy for a given prompt is higher than the corresponding average under the reference policy (Fig. S10). Furthermore, we see that we are able to sample a diverse set of molecules that are chemically similar to the prompt molecule, and also chemically valid (see Fig. S11, Table S8). We repeat the experiment with a related objective of generating molecules similar to the prompt molecule with a high LogP instead and again observe that we increase the per-prompt average LogP under the optimized policy relative to the reference policy without degrading sample diversity and validity. For both of these experiments, we required regularization to the reference policy ($\gamma > 0$). With no regularization, the aligned generator would almost exclusively sample sequences that were chemically invalid ($< 25\%$ chemical validity). Finally, we note that the increases in QED and LogP in Fig. S10 are smaller relative to the increases in Fig. 2 because the samples are now conditioned to remain proximal to the prompt molecule, which restricts the chemical space that can be explored.

### 4.1.3 Unprompted molecular alignment on protein-ligand docking oracles

| | GSK3$\beta$ top-100 | | JNK3 top-100 | |
| --- | --- | --- | --- | --- |
| | mean score | IntDiv | mean score | IntDiv |
| ERA | $0.996 \pm 0.000$ | $\mathbf{0.219 \pm 0.002}$ | $\mathbf{0.987 \pm 0.001}$ | $\mathbf{0.264 \pm 0.005}$ |
| MolRL-MGPT | $\mathbf{1.000 \pm 0.000}$ | $0.362 \pm 0.015$ | $0.961 \pm 0.010$ | $0.372 \pm 0.025$ |
| GFlowNet | $0.649 \pm 0.072$ | $0.715 \pm 0.104$ | $0.437 \pm 0.219$ | $0.716 \pm 0.145$ |
| GraphGA | $0.919 \pm 0.016$ | $0.365 \pm 0.024$ | $0.875 \pm 0.025$ | $0.380 \pm 0.015$ |
| JT-VAE | $0.235 \pm 0.083$ | $0.770 \pm 0.067$ | $0.159 \pm 0.040$ | $0.781 \pm 0.127$ |
| REINVENT | $0.965 \pm 0.011$ | $0.308 \pm 0.035$ | $0.942 \pm 0.019$ | $0.368 \pm 0.021$ |

Table 1: Mean scores and internal diversities (IntDiv) of experiments on GSK3$\beta$ and JNK3 tasks averaged across 5 random seeds. For each task, 20k molecules were sampled, and metrics were computed on top-100 scoring *valid, novel and unique* molecules filtered from the initial 20K samples (i.e. molecules not in dataset and molecules not previously sampled). Compared to state-of-the-art methods, ERA samples more diverse molecules with higher predicted docking scores. Results for compared methods are reproduced from Hu et al. [2023].

We next investigate the performance of ERA in designing compounds that have high predicted docking scores for the kinases JNK3 and GSK3$\beta$. For each of these targets, we use an *in silico* oracle that predicts docking scores, ranging from 0 to 1, where a higher value corresponds to stronger predicted score Sun et al. [2017]. Using only data from ChemBL, we first carry out a short supervised fine-tuning step on all molecules in ChemBL with an oracle score above 0.5 (7386 molecules for JNK3 and 43381 for GSK3$\beta$). Using this fine-tuned model as our reference policy, we then carry out alignnment using ERA ($\beta$=100 and $\gamma$=0), where we use a comparably high $\beta$ to target molecules with high activity. As with the QED alignment runs in Section 4.1.1, we define the energy for this task as the negative logarithm of the oracle score.

From the aligned models, we sample 20000 molecules (see Fig. S12) and tabulate metrics of the top-100 performing molecules (see Table 1). We note that the molecules in the top-100 are both *novel* and *unique* after filtering to exclude any molecules that are present in the ChemBL dataset and any repeated molecules. For GSK3$\beta$, our mean score is marginally lower than the best performing method but the diversity in sampled molecules is significantly higher (i.e. lower IntDiv). For JNK3 our mean score is significantly higher than the best performing method *and* the diversity in sampled molecules is higher than any method. The inference costs are notably low for our approach; sampling 20000 molecules and filtering takes only minutes on a single GPU.

We additionally measure sample efficiency using the top-10 AUC metric Gao et al. [2022], Bou et al. [2024], which is the area under the curve (AUC) of the mean property value of the top-10 performing molecules versus the number of oracle calls (see Fig. S13 and Table S9). We likewise only include novel, unique, and valid molecules in this analysis. We observe that we are able to generate novel and unique high-scoring molecules, with high sample efficiency especially in comparison to existing state-of-the-art methods such as REINVENT, GraphGA, PPO, and PPOD Gao et al. [2022], Bou et al. [2024]. Ultimately, high sample efficiency is crucial in settings where evaluation is expensive, which will generally be true for most real-world chemical and biological tasks (e.g. wet-lab experiment). Finally, we also perform Glide Standard Precision Friesner et al. [2004] docking on the top-scoring molecules according to the oracles (score of 1.0) against their respective receptors. We observe that the diverse set of sampled molecules exhibit chemically plausible docked poses obtained from a physics-based docking approach (Fig. 4).

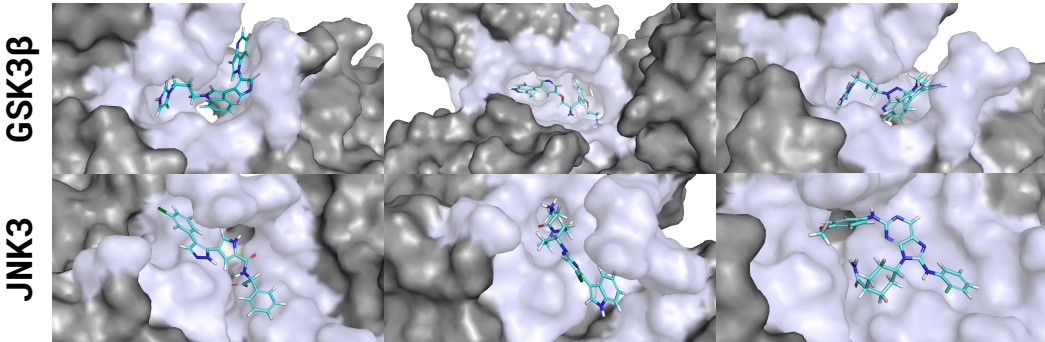

Figure 4: Visualization of three generated ligands docked against the GSK3$\beta$ kinase target (top) and three generated ligands docked against the JNK3 kinase target (bottom). In each case, these were the three molecules with the best (most negative) Glide Standard Precision docking scores and oracle scores of 1.0.

## 4.2 Directed evolution of proteins with ERA

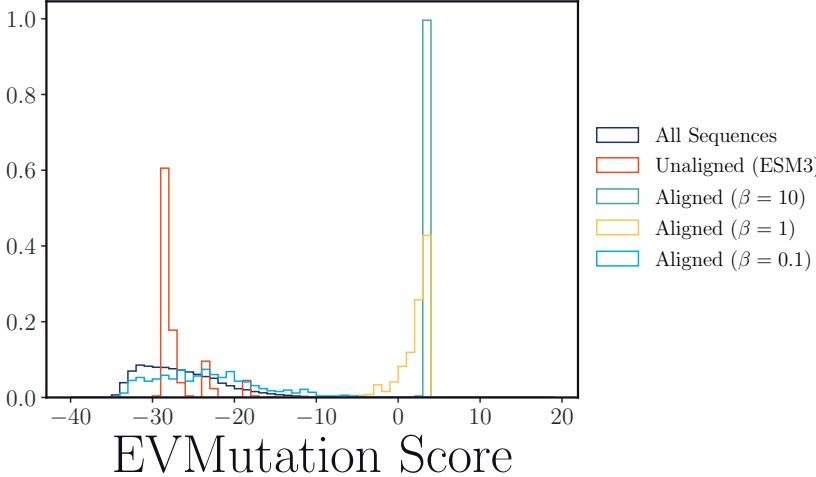

Figure 5: Alignment of ESM3-1.4B with $\beta$=0, 0.1, 1.0, 10.0 and $\gamma$=0.001 on the task of maximizing EVmutation score. Positions 182, 183, 184, and 186 of the TrpB parent sequence were masked and ESM3-1.4B predicted amino acids at those sites. The distribution of the EVmutation scores for generated sequences shifts significantly as $\beta$ is increased.

We also consider the performance of ERA in a large-molecule setting, namely ML-guided directed evolution of proteins. Directed evolution campaigns aim to optimize a protein sequence toward some desired property of interest via iterative mutagenesis, library screening, and selection of best variants

Wang et al. [2021]. This has become a widely used methodology in protein engineering but comes with key limitations. The inherently iterative nature of directed evolution campaigns can lead to costly and time-consuming experimental campaigns, and meaningfully understanding the effects of protein mutations on protein activity can often be difficult. These challenges have led to the application of machine learning methods to more efficiently guide directed evolution campaigns Yang et al. [2024] Shanker et al. [2024]. Given the success of ERA in guiding the optimization of small-molecules using a SMILES language representation, we examined whether ERA could be used to optimize large protein molecules using a protein language (i.e. primary sequence) representation.

There has been significant recent effort to design and train large protein language models (PLMs) Lin et al. [2023], Hayes et al. [2024]. Furthermore, these models have demonstrated remarkable capabilities across a number of protein tasks Widatalla et al. [2024], Shanker et al. [2024]. As such, we decided to use the state-of-the-art ESM3-1.4B Hayes et al. [2024] as our pretrained model, for which we carried out alignment using ERA. Despite the multimodal nature of ESM3, here, we only focus on generating primary-sequence-based representations of proteins.

We consider directed evolution of the $\beta$-subunit of tryptophansynthase (TrpB) from *Thermotoga maritima*, an enzyme that catalyzes tryptophan production Buller et al. [2015]. Here, we seek to evolve the protein to increase its evolutionary fitness. In this work, we do not have access to experimental validation and so we evaluate the fitness of sequences using the computationally evaluable EVmutation score, an oracle that is predictive of a variant sequence's performance relative to the parent sequence in its native function Hopf et al. [2017].

As in other directed evolution campaigns for the TrpB protein Yang et al. [2023], we consider mutating four different sites to one of the 20 standard amino acids. We randomly sampled 512 mutated sequences, emulating a random mutagenesis experiment. Using ESM3-1.4B as our reference model, we carry out alignment using ERA with various $\beta = (0.1, 1.0, 10.0)$ and $\gamma = 0.001$ and plot the results in Fig. 5 (see Appendix D for more details). We observe that with higher $\beta$, we are able to sample mutants with the highest possible EVmutation score in a single round of alignment. These results are promising for the application of ERA in directed evolution campaigns and future work will focus on the guidance of wet-lab directed evolution campaigns in conjunctions with multi-round, on-policy ERA.

# 5    Conclusions and Limitations

This paper introduces energy rank alignment, a simple and effective algorithm for policy optimization with an explicit reward model. We find that ERA is stable without extensive hyperparameter tuning, and sufficiently general to successfully align application-specific transformers for chemical search problems and protein language models. The algorithm exhibits strong performance with a variety of reward models, even ones with relatively weak signal. We analyze the minimizers of the ERA objective and find that they differ from the minimizers of popular policy alignment algorithms DPO and PPO in an important way: unlike PPO, the strength of regularization to the reference policy that we add is controlled by a parameter $\gamma$, while the entropy of the target distribution is independently tuned by a distinct parameter $\beta$. This means that we can avoid greedy policies by keeping $\beta$ small—amplifying fluctuations around the optimum of the reward model ($-U$)—while reducing the influence of the reference policy by taking $\gamma$ small. Our objective leads to easily interpretable sample-wise gradients which highlight the importance of a reward model relative to DPO in the sampled objective.

**Limitations:**    First, our approach requires a reward model, which can be difficult to train or design, especially for complex tasks. While we observed that ERA makes an appreciable impact even with weak supervision, this sort of proxy may not be available for more complex tasks. For example, optimizing small molecules for high binding affinity to a target protein would require expensive and noisy evaluations of a reward model, which likely limits the scope of molecular design to problems where the reward can be computed somewhat efficiently. A second limitation of our present work is that we do not train the molecular transformer to favor synthetic accessibility nor do we explicitly seek to obtain molecules that are easily synthesized experimentally. There are models that seek to evaluate synthesizability computationally that could be used in our rewards, which we plan to explore in future work Coley et al. [2018].

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

# A Detailed Theoretical Analysis

**Set-up, notation, and assumptions**   Let $\mathcal{X}$ and $\mathcal{Y}$ be discrete spaces; each element of one of these spaces is a finite-length sequence of tokens within a fixed dictionary on which an autoregressive generative model is trained. The resulting models yield "policies", which are conditional probability distributions $\pi(\cdot|\boldsymbol{x}) \in \mathcal{P}(\mathcal{Y})$ for each $\boldsymbol{x} \in \mathcal{X}$. Throughout, we assume that our policies have full support on $\mathcal{Y}$ for each $\boldsymbol{x}$, meaning that $\inf_{\boldsymbol{y},\boldsymbol{x}\in\mathcal{Y}\times\mathcal{X}} \pi(\boldsymbol{y}|\boldsymbol{x}) > 0$. Because the spaces are discrete, we make no strong restrictions on the regularity or coerciveness of the reward model $-U : \mathcal{X} \times \mathcal{Y} \to \mathbb{R}$. The only requirement to ensure the existence of an optimal probability distribution is that $\sup_{\boldsymbol{x},\boldsymbol{y}\times\mathcal{X}\times\mathcal{Y}} |e^{-U(\boldsymbol{x},\boldsymbol{y})}| < +\infty$, which maintains full support of the distribution. Though it plays little role in theoretical analysis, we also denote by $\nu \in \mathcal{P}(\mathcal{X})$ the probability distribution over the prompts $\boldsymbol{x}$.

**Goals of the analysis presented here**   The main purpose of this section is to establish that globally minimizing the loss (9) yields a global minimizer of the regularized policy objective (2). A secondary goal is to clearly articulate the theoretical advantages of ERA compared with PPO and DPO.

To understand the ERA loss function and its connection to the entropy regularized objective (2), we first establish that the minimizer of (19) are of the form (5). We first define the notion of equivalence precisely.

**Definition A.1** *The conditional probability measures $\pi(\cdot|\boldsymbol{x})$ and $\pi'(\cdot|\boldsymbol{x})$ in $\mathcal{P}(\mathcal{Y})$ are conditionally equivalent if $\forall \boldsymbol{x} \in \mathcal{X}$, $\pi$ and $\pi'$ are such that $\sup_{\boldsymbol{y}\in\mathcal{Y}} |\pi(\boldsymbol{y}|\boldsymbol{x}) - \pi'(\boldsymbol{y}|\boldsymbol{x})| = 0$.*

This is a strong form of equivalence for probability measures, but it is appropriate on the discrete spaces $\mathcal{X}$ and $\mathcal{Y}$ we consider here. For more general continuous spaces, one could relax this condition to weak equivalence of the conditional measures. We use this notion to emphasize that a shift of the distribution of the "prompts" $\boldsymbol{x} \in \mathcal{X}$, which we denote $\nu \in \mathcal{P}(\mathcal{X})$, does not impact conditional equivalence and hence establishes an equivalence class of conditional probability measures that minimize (2).

**Lemma A.1** *If $\pi$ is conditionally equivalent to $\pi'$, then $\pi'_g(\cdot|\boldsymbol{x}) \propto \pi'(\cdot|\boldsymbol{x})e^{g(\boldsymbol{x})}$ is conditionally equivalent to $\pi$ for all functions $g : \mathcal{X} \to \mathbb{R}$ such that $\sup_{\boldsymbol{x}\in\mathcal{X}} |e^{g(\boldsymbol{x})}| < +\infty$.*

Assume that $\pi'$ is a normalized probability distribution. This requires that,

$$Z'(\boldsymbol{x}) = \sum_{\boldsymbol{y}\in\mathcal{Y}} \pi'(\boldsymbol{y}|\boldsymbol{x}) = 1. \tag{14}$$

If $g$ is such that

$$Z'_g(\boldsymbol{x}) = \sum_{\boldsymbol{y}\in\mathcal{Y}} \pi'(\boldsymbol{y}|\boldsymbol{x})e^{g(\boldsymbol{x})} \neq 1, \tag{15}$$

then the normalized policy $\pi'_g$ is clearly defined by

$$\frac{1}{Z'_g(\boldsymbol{x})} \pi'(\boldsymbol{y}|\boldsymbol{x})e^{g(\boldsymbol{x})} \equiv \pi'(\boldsymbol{y}|\boldsymbol{x}), \tag{16}$$

because $Z'_g(\boldsymbol{x}) = e^{g(\boldsymbol{x})}$. By the assumption that $\sup_{\boldsymbol{x}\in\mathcal{X}} |e^{g(\boldsymbol{x})}| < +\infty$, all terms in these calculations remain finite.

Using Lemma A.1 it is straightforward to prove the result in Proposition 3.2. For completeness, we re-state that result here and refer the reader to Appendix 3 for the complete argument.

**Proposition A.2** *Suppose $\pi(\cdot|\boldsymbol{x}) \in \mathcal{P}(\mathcal{Y})$ and that $\mathrm{supp}(\pi) = \mathrm{supp}(\pi_{\mathrm{ref}})$. Let $\beta > 0$, $\gamma \geq 0$ and that the reward model is such that $\sup_{\boldsymbol{x},\boldsymbol{y}\in\mathcal{X}\times\mathcal{Y}} |e^{-U(\boldsymbol{x},\boldsymbol{y})}| < +\infty$. Then, the minimizer of $\mathcal{L}^{\mathrm{ERA}}$ is conditionally equivalent to $\pi_\star$.*

This proposition establishes that a policy minimizing the objective

$$\mathcal{L}^{\mathrm{ERA}}(\pi_{\boldsymbol{\theta}}) = \mathbb{E}_{x\sim\mathcal{D}}\mathbb{E}_{\boldsymbol{y},\boldsymbol{y}'\sim\pi_{\mathrm{ref}}(\cdot|\boldsymbol{x})}D_{\mathrm{KL}}^{(\boldsymbol{y},\boldsymbol{y}')}(p_{\beta}|p_{\boldsymbol{\theta}});$$

$$p_{\boldsymbol{\theta}} := \sigma\left(\log\frac{\pi_{\boldsymbol{\theta}}(\boldsymbol{y}|\boldsymbol{x})}{\pi_{\boldsymbol{\theta}}(\boldsymbol{y}'|\boldsymbol{x})}\right) \tag{17}$$

$$p_{\gamma} := \sigma\left(\frac{\beta}{1+\gamma}\left[(U(\boldsymbol{x},\boldsymbol{y}') - U(\boldsymbol{x},\boldsymbol{y})) + \beta^{-1}\gamma\log\frac{\pi_{\mathrm{ref}}(\boldsymbol{y}|\boldsymbol{x})}{\pi_{\mathrm{ref}}(\boldsymbol{y}'|\boldsymbol{x})}\right]\right),$$

has the form

$$\pi_{\star}(\boldsymbol{y}|\boldsymbol{x}) = Z^{-1}(\boldsymbol{x})\exp\left[-\frac{\beta}{1+\gamma}\big(U(\boldsymbol{x},\boldsymbol{y}) - \beta^{-1}\gamma\log\pi_{\mathrm{ref}}(\boldsymbol{y}|\boldsymbol{x})\big)\right]. \tag{18}$$

We do not, however, prove that gradient descent of $\boldsymbol{\theta}$ on (17) converges to the global minimizer (18) because such an argument requires additional assumptions about the parametric class of policies and the convexity of the objective with respect to the parameters, neither of which are straightforward to establish.

### A.1 Loss functions for $\pi_{\boldsymbol{\theta}}$:

Proximal Policy Optimization (PPO) optimizes an indirect, proximal objective to minimize an objective closely related to (2) (cf. Appendix A). Direct Preference Optimization (DPO) treats the negative reward function $U$ implicitly and directly maximizes the likelihood of $p(\boldsymbol{y}\succ\boldsymbol{y}'|\boldsymbol{x})$. Our objectives differ from both approaches: like DPO, we directly optimize the policy using an explicit, gradient-based objective, but, in contrast, we use a reward function directly in our objective. The losses we build are thus amenable to both offline (samples from $\pi_{\mathrm{ref}}$) and online (samples from $\pi_{\boldsymbol{\theta}}$) policy alignment, as explained below. Choosing to optimize the objective online has been shown to have important consequences on performance Tajwar et al. [2024], though we focus here on the setting where samples are drawn offline.

We directly optimize the Kullback-Leibler divergence between the entropy-regularized preference distribution $p_{\gamma}(\boldsymbol{y}\succ\boldsymbol{y}'|\boldsymbol{x})$ and the corresponding parametric preference distribution $p_{\boldsymbol{\theta}}(\boldsymbol{y}\succ\boldsymbol{y}'|\boldsymbol{x})$. Explicitly, using the fact that conditional preference distribution is normalized, we obtain

$$D_{\mathrm{KL}}^{(\boldsymbol{y},\boldsymbol{y}')}(p_{\gamma}|p_{\boldsymbol{\theta}}) = p_{\gamma}(\boldsymbol{y}\succ\boldsymbol{y}'|\boldsymbol{x})\log\frac{p_{\gamma}(\boldsymbol{y}\succ\boldsymbol{y}'|\boldsymbol{x})}{p_{\boldsymbol{\theta}}(\boldsymbol{y}\succ\boldsymbol{y}'|\boldsymbol{x})} + p_{\gamma}(\boldsymbol{y}'\succ\boldsymbol{y}|\boldsymbol{x})\log\frac{p_{\gamma}(\boldsymbol{y}'\succ\boldsymbol{y}|\boldsymbol{x})}{p_{\boldsymbol{\theta}}(\boldsymbol{y}'\succ\boldsymbol{y}|\boldsymbol{x})},$$

$$= p_{\gamma}(\boldsymbol{y}\succ\boldsymbol{y}'|\boldsymbol{x})\log\frac{p_{\gamma}(\boldsymbol{y}\succ\boldsymbol{y}'|\boldsymbol{x})}{p_{\boldsymbol{\theta}}(\boldsymbol{y}\succ\boldsymbol{y}'|\boldsymbol{x})} + \big(1 - p_{\gamma}(\boldsymbol{y}\succ\boldsymbol{y}'|\boldsymbol{x})\big)\log\frac{1 - p_{\gamma}(\boldsymbol{y}\succ\boldsymbol{y}'|\boldsymbol{x})}{1 - p_{\boldsymbol{\theta}}(\boldsymbol{y}\succ\boldsymbol{y}'|\boldsymbol{x})}, \tag{19}$$

where

$$p_{\gamma} := \sigma\left(\frac{\beta}{1+\gamma}\left[(U(\boldsymbol{x},\boldsymbol{y}') - U(\boldsymbol{x},\boldsymbol{y})) + \beta^{-1}\gamma\log\frac{\pi_{\mathrm{ref}}(\boldsymbol{y}|\boldsymbol{x})}{\pi_{\mathrm{ref}}(\boldsymbol{y}'|\boldsymbol{x})}\right]\right). \tag{20}$$

This quantity is a well-defined KL divergence and is hence non-negative; the quantity vanishes when $p_{\gamma} = p_{\boldsymbol{\theta}}$ on the observations $\boldsymbol{y},\boldsymbol{y}'$. Furthermore, with access to an explicit reward model, all terms in (19) can be computed directly and

$$p_{\boldsymbol{\theta}}(\boldsymbol{y}\succ\boldsymbol{y}'|\boldsymbol{x}') = \frac{\pi_{\boldsymbol{\theta}}(\boldsymbol{y}|\boldsymbol{x})}{\pi_{\boldsymbol{\theta}}(\boldsymbol{y}|\boldsymbol{x}) + \pi_{\boldsymbol{\theta}}(\boldsymbol{y}'|\boldsymbol{x})} = \sigma\left(\log\frac{\pi_{\boldsymbol{\theta}}(\boldsymbol{y}|\boldsymbol{x})}{\pi_{\boldsymbol{\theta}}(\boldsymbol{y}'|\boldsymbol{x})}\right). \tag{21}$$

To obtain a minimizer of the regularized objective defined in (2) we optimize

$$\mathcal{L}^{\mathrm{ERA}}(\pi_{\boldsymbol{\theta}}) = \mathbb{E}_{x\sim\mathcal{D}}\mathbb{E}_{\boldsymbol{y},\boldsymbol{y}'\sim\pi_{\mathrm{ref}}(\cdot|\boldsymbol{x})}D_{\mathrm{KL}}^{(\boldsymbol{y},\boldsymbol{y}')}(p_{\gamma}|p_{\boldsymbol{\theta}}); \tag{22}$$

If the current policy overlaps with the target preference distribution, it may be useful to sample directly from the partially aligned policy, i.e., to use the "on-policy" formulation,

$$\mathcal{L}_{\mathrm{on}}^{\mathrm{ERA}}(\pi_{\boldsymbol{\theta}}) = \mathbb{E}_{\boldsymbol{x}\sim\mathcal{D}}\mathbb{E}_{\boldsymbol{y},\boldsymbol{y}'\sim\pi_{\boldsymbol{\theta}}(\boldsymbol{y}|\boldsymbol{x})}D_{\mathrm{KL}}^{(\boldsymbol{y},\boldsymbol{y}')}(p_{\gamma}|p_{\boldsymbol{\theta}}) \tag{23}$$

instead of (9). One issue that arises with this scheme is differentiation with respect to the parameters of the policy $\boldsymbol{\theta}$ because $\boldsymbol{y}$ and $\boldsymbol{y}'$ are decoded into discrete tokens, an operation that is not differentiable. To remedy this, we importance sample with a reference policy

$$\mathcal{L}_{\mathrm{on}}^{\mathrm{ERA}}(\pi_{\boldsymbol{\theta}}) = \mathbb{E}_{\boldsymbol{x}\sim\mathcal{D}}\mathbb{E}_{\boldsymbol{y},\boldsymbol{y}'\sim\pi_{\mathrm{ref}}(\boldsymbol{y}|\boldsymbol{x})}\frac{\pi_{\boldsymbol{\theta}}(\boldsymbol{y}|\boldsymbol{x})\pi_{\boldsymbol{\theta}}(\boldsymbol{y}'|\boldsymbol{x})}{\pi_{\mathrm{ref}}(\boldsymbol{y}|\boldsymbol{x})\pi_{\mathrm{ref}}(\boldsymbol{y}'|\boldsymbol{x})}D_{\mathrm{KL}}^{(\boldsymbol{y},\boldsymbol{y}')}(p_{\gamma}|p_{\boldsymbol{\theta}}). \tag{24}$$

This reweighting is straightforward and the importance weights should generally be appreciable, especially early in training when $\pi_{\boldsymbol{\theta}}$ has not drifted far from $\pi_{\text{ref}}$. It is, of course, also natural to iteratively update $\pi_{\boldsymbol{\theta}}$ using a previous iterate as the reference policy. In this work, we only use (9) as an objective and leave the on-policy objectives to future work.

## A.2 Gradients of $\mathcal{L}^{\text{ERA}}$.

One advantage of the ERA framework is that the objective is amenable to direct, gradient-based optimization. We remark that establishing global convergence for the optimization of $\boldsymbol{\theta}$ using (9) requires establishing convexity with respect to the parameters, which is not obviously the case for our objective, nor those used in PPO and DPO. However, one can still glean some insight into the optimization by examining the gradients on a samplewise basis. Using the compact notation $p_{\boldsymbol{\theta}}(\boldsymbol{y} \succ \boldsymbol{y}'|\boldsymbol{x}) \equiv \sigma_{\boldsymbol{\theta}}$ and $p_{\gamma}(\boldsymbol{y} \succ \boldsymbol{y}'|\boldsymbol{x}) \equiv \sigma_{\star}$,

$$\nabla_{\boldsymbol{\theta}} \mathcal{L}^{\text{ERA}} = \mathbb{E}_{\boldsymbol{x} \sim \mathcal{D}} \mathbb{E}_{\boldsymbol{y}, \boldsymbol{y}' \sim \pi_{\text{ref}}} \left( \frac{1 - \sigma_{\star}}{1 - \sigma_{\boldsymbol{\theta}}} - \frac{\sigma_{\star}}{\sigma_{\boldsymbol{\theta}}} \right) \nabla_{\boldsymbol{\theta}} \sigma_{\boldsymbol{\theta}}. \tag{25}$$

The gradient is straightforward to interpret on a particular pair $\boldsymbol{y}, \boldsymbol{y}'$: if $p_{\boldsymbol{\theta}}(\boldsymbol{y} \succ \boldsymbol{y}'|\boldsymbol{x})$ is larger than $p_{\gamma}(\boldsymbol{y} \succ \boldsymbol{y}'|\boldsymbol{x})$ then the preference gradient is positive and gradient descent lowers the probability that $\boldsymbol{y} \succ \boldsymbol{y}'$. The opposite occurs whenever $p_{\boldsymbol{\theta}}(\boldsymbol{y} \succ \boldsymbol{y}'|\boldsymbol{x})$ is smaller than $p_{\gamma}(\boldsymbol{y} \succ \boldsymbol{y}'|\boldsymbol{x})$. The magnitude of the gradient is scaled by the degree of misspecification of the preference probability.

This calculation highlights one key difference between the approach we use and DPO. When the data only contains one observation of $\boldsymbol{y} \succ \boldsymbol{y}'$ for a given $\boldsymbol{x}$, the DPO objective's implicit reward model assigns zero probability to $\boldsymbol{y}' \succ \boldsymbol{y}$. This pushes the policy towards extremal values, which can lead to undesired behavior, as discussed in Azar et al. [2023]. In our formulation, this behavior occurs only when the reward model assigns an energy of $\pm\infty$, which is prohibited by construction in most tasks. We further discuss differences between ERA and DPO in Appendix A.4.

## A.3 Comparison with PPO Objective

The free energy functional for a policy under the energy rank alignment framework can be written as an expectation

$$J_{\text{ERA}}[\pi] = \mathbb{E}_{\boldsymbol{x} \sim \nu} \left[ \int U(\boldsymbol{x}, \boldsymbol{y}) \mathrm{d}\pi(\boldsymbol{y}|\boldsymbol{x}) + \beta^{-1} \int (1 + \gamma) \log \pi(\boldsymbol{y}|\boldsymbol{x}) - \gamma \log(\pi_{\text{ref}}(\boldsymbol{y}|\boldsymbol{x}) \mathrm{d}\pi(\boldsymbol{y}|\boldsymbol{x}) \right], \tag{26}$$

involving an energetic term and an entropic term. The additional regularization acts as an effective energetic bias. Solving for the extremum of this functional by setting Fréchet derivative with respect to $\pi$ equal to zero, one obtains the formal solution (18) for the minimizer. This objective differs from the regularized reward loss conventionally used for PPO,

$$\begin{aligned} J_{\text{PPO}}(\pi) &= \mathbb{E}_{\boldsymbol{x}} \left[ \int U(\boldsymbol{x}, \boldsymbol{y}) \mathrm{d}\pi(\boldsymbol{y}|\boldsymbol{x}) + \gamma \beta^{-1} \int \log \frac{\pi(\boldsymbol{y}|\boldsymbol{x})}{\pi_{\text{ref}}(\boldsymbol{y}|\boldsymbol{x})} \mathrm{d}\pi(\boldsymbol{y}|\boldsymbol{x}) \right], \\ &= \mathbb{E}_{\boldsymbol{x}} \left[ \int U(\boldsymbol{x}, \boldsymbol{y}) \mathrm{d}\pi(\boldsymbol{y}|\boldsymbol{x}) + \gamma \beta^{-1} D_{\text{KL}} \big( \pi(\cdot|\boldsymbol{x}) | \pi_{\text{ref}}(\cdot|\boldsymbol{x}) \big) \right]. \end{aligned} \tag{27}$$

The minimizer of the PPO objective (27) is also a Gibbs-Boltzmann measure, explicitly,

$$\pi_{\star}^{(\text{PPO})} \propto \exp \left[ -\frac{\beta}{\gamma} U(\boldsymbol{x}, \boldsymbol{y}) + \log \pi_{\text{ref}}(\boldsymbol{y}|\boldsymbol{x}) \right]. \tag{28}$$

Here, the KL-regularization corresponds to an energy shift, as in our objective, but there is no limit in which the ideal distribution $\pi \propto e^{-\beta U}$ is obtained for the PPO objective. This is in stark contrast to our approach, which recovers the ideal distribution as $\gamma \to 0$. Furthermore, while our approach allows for a direct gradient-based optimization using (17), PPO is implemented using an actor-critic framework that is difficult to tune Rafailov et al. [2023], Casper et al. [2023]. Finally, we emphasize that for ERA in the $\gamma \to 0$, finite $\beta > 0$, the distribution has positive entropy and is not manifestly mode-seeking; there can still be appreciable fluctuations in the output. Eliminating the effect of regularization in (28), on the other hand, requires taking $\beta/\gamma \to \infty$, which eliminates fluctuations in the distribution.

## A.4 Comparison with DPO Objective

The DPO approach also seeks to optimize the objective (27). The algorithm does so by first using (28) to define an implicit reward model by solving for the $U$ that reflects the observed preference probabilities. This elegant idea has had a significant impact and has already been deployed in state-of-the-art models AI@Meta [2024]. In many cases, the observed preference probabilities will be sampled and only perhaps only one observation of $\boldsymbol{y} \succ \boldsymbol{y}'$ will be available for each $\boldsymbol{x}$ in the dataset. When the preference dataset only has one observation $\boldsymbol{y} \succ \boldsymbol{y}'$ per prompt $\boldsymbol{x}$, the optimal policy requires that

$$\pi_\star^{\mathrm{DPO}}(\boldsymbol{y}|\boldsymbol{x}) = 1 \quad \text{and} \quad \pi_\star^{\mathrm{DPO}}(\boldsymbol{y}'|\boldsymbol{x}) = 0. \tag{29}$$

The sampled gradients of the objective used for DPO are proportional to the implicit reward discrepancy,

$$\nabla_{\boldsymbol{\theta}} \hat{\mathcal{L}}^{\mathrm{DPO}}(\boldsymbol{y}, \boldsymbol{y}', \boldsymbol{x}) = \sigma \left( \beta^{-1}\gamma \left[ \log \frac{\pi_{\boldsymbol{\theta}}(\boldsymbol{y}'|\boldsymbol{x})}{\pi_{\mathrm{ref}}(\boldsymbol{y}'|\boldsymbol{x})} - \log \frac{\pi_{\boldsymbol{\theta}}(\boldsymbol{y}|\boldsymbol{x})}{\pi_{\mathrm{ref}}(\boldsymbol{y}|\boldsymbol{x})} \right] \right) \nabla_{\boldsymbol{\theta}} \log \frac{\pi_{\boldsymbol{\theta}}(\boldsymbol{y}|\boldsymbol{x})}{\pi_{\boldsymbol{\theta}}(\boldsymbol{y}'|\boldsymbol{x})}, \tag{30}$$

which when $\pi_{\boldsymbol{\theta}}(\boldsymbol{y}'|\boldsymbol{x}) \to 0$, could lead to instability as $-\log \pi_{\boldsymbol{\theta}}(\boldsymbol{y}'|\boldsymbol{x}) \to \infty$. On the other hand, the ERA gradients are scaled by the relative preference discrepancy,

$$\nabla_{\boldsymbol{\theta}} \mathcal{L}^{\mathrm{ERA}}(\boldsymbol{y}, \boldsymbol{y}', \boldsymbol{x}) = \left( \frac{1 - \sigma_\star(\boldsymbol{y} \succ \boldsymbol{y}'|\boldsymbol{x})}{1 - \sigma_{\boldsymbol{\theta}}(\boldsymbol{y} \succ \boldsymbol{y}'|\boldsymbol{x})} - \frac{\sigma_\star(\boldsymbol{y} \succ \boldsymbol{y}'|\boldsymbol{x})}{\sigma_{\boldsymbol{\theta}}(\boldsymbol{y} \succ \boldsymbol{y}'|\boldsymbol{x})} \right) \nabla_{\boldsymbol{\theta}} \sigma_{\boldsymbol{\theta}}(\boldsymbol{y} \succ \boldsymbol{y}'|\boldsymbol{x}). \tag{31}$$

The advantage of a reward model becomes apparent because

$$\sigma_\star(\boldsymbol{y} \succ \boldsymbol{y}'|\boldsymbol{x}) = p_\gamma(\boldsymbol{y} \succ \boldsymbol{y}'|\boldsymbol{x}) = \sigma \left( \frac{\beta}{1+\gamma} \left[ (U(\boldsymbol{x}, \boldsymbol{y}') - U(\boldsymbol{x}, \boldsymbol{y})) + \beta^{-1}\gamma \log \frac{\pi_{\mathrm{ref}}(\boldsymbol{y}|\boldsymbol{x})}{\pi_{\mathrm{ref}}(\boldsymbol{y}'|\boldsymbol{x})} \right] \right) \tag{32}$$

and hence the optimum of $\mathcal{L}^{\mathrm{ERA}}$ will not lead to policies in which $\mathrm{supp}(\pi_{\boldsymbol{\theta}})$ degrades unless the energy becomes infinite. Choosing an appropriate reward model, hence, gives the flexibility to control instability if it becomes problematic.

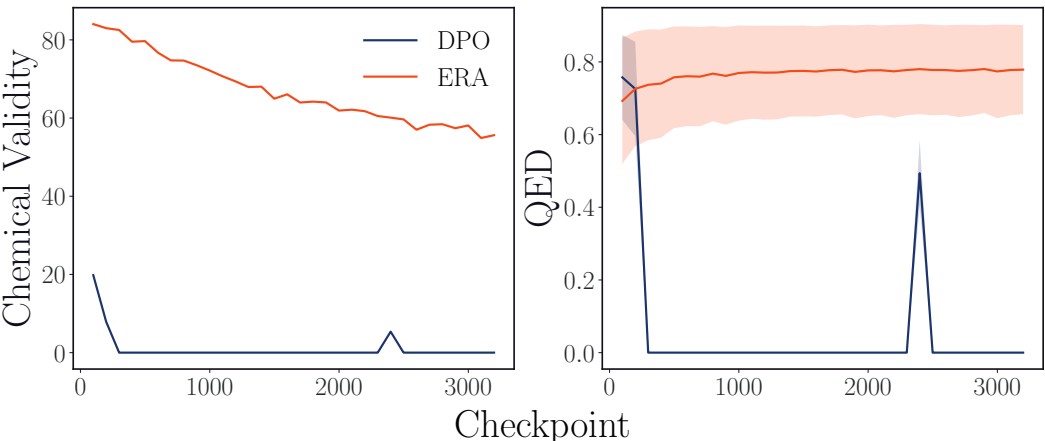

Figure S6: Comparison of DPO ($\beta_{\mathrm{DPO}}$=0.1) and ERA ($\beta = 20.0$ and $\gamma = 0.0$) on the task of maximizing QED (Section 4.1.1) a) Chemical validity degrades significantly for DPO and slightly degrades for ERA as a function of checkpoints. b) At each checkpoint, 20k molecules are sampled and mean QED of valid molecules plotted with shading corresponding to 1 standard deviation. Both DPO and ERA alignment runs were trained with the same hyperparameters and dataset, with checkpoints saved every 100 epochs.

We carry out an online evaluation of DPO and ERA on the task of generating small-molecules with high QED, as in Figure 2. We align DPO using the same dataset and hyperparameters as ERA with $\beta_{DPO} = 0.1$ and train for thousands of checkpoints (over 72 GPU-hours). We load intermediate checkpoints for both the DPO and ERA ($\beta_{ERA} = 20.0$, $\gamma_{ERA} = 0.0$) runs and carry out inference (Figure S6). We observe that at the first saved checkpoint of the DPO alignment run, the model generates molecules with high QED scores but with low validity (~20%). However, upon further

training, the chemical validity of further checkpoints drops to 0% for the remaining runs, despite the overall DPO training and validation losses still dropping.

With ERA, we see that we are able to similarly sample high QED small-molecules with reasonably high chemical validity ($\sim$85%). While the validity does drop over subsequent checkpoints, it does not do so precipitously. Moreover, the ERA-based alignment had no regularization ($\gamma = 0$), and in the main text, we document how having non-zero $\gamma$ can enable increases in chemical validity (Figure S10). Finally, we note that we did not extensively tune the hyperparameters for DPO, and it is possible that a different set of hyperparameters would elicit a more desired outcome; however, the lack of meaningful regularization in DPO Azar et al. [2023] and its performance degradation in online metrics has been well-documented Chen et al. [2024].

## B  ERA implementation

Implementing energy rank alignment is straightforward to implement within existing code bases. We provide sample PyTorch code for the ERA loss function below.

```
import torch.nn as nn
from torch.nn.functional import logsigmoid

def era_loss(pi_logps_1, pi_logps_2,
             ref_logps_1, ref_logps_2,
             energies_1, energies_2,
             beta, gamma):
    """
    pi_logps_1: logprob under policys model of first sequence in pair (B,)
    pi_logps_2: logprob under policys model of second sequence in pair (B,)
    ref_logps_1: logprob under reference model of first sequence in pair (B,)
    ref_logps_2: logprob under reference model of second sequence in pair (B,)
    energies_1: energies of first sequence in pair (B,)
    energies_2: energies of second sequence in pair (B,)
    beta: inverse temperature
    gamma: regularization controlling strength of KL penalty
    """
    beta_prime = (beta / (1 + gamma))
    gamma_prime = (gamma / (1 + gamma))

    logp = logsigmoid(policy_logps_y2 - policy_logps_y1)
    logp_prime = logsigmoid(policy_logps_y1 - policy_logps_y2)

    logp_star = logsigmoid(-beta_prime * (energies_y2 - energies_y1)
                            + gamma_prime * (ref_logps_y2 - ref_logps_y1))
    logp_star_prime = logsigmoid(-beta_prime * (energies_y1 - energies_y2)
                            + gamma_prime * (ref_logps_y1 - ref_logps_y2))

    era_loss = (torch.exp(logp_star) * (logp_star - logp)
            + torch.exp(logp_star_prime) * (logp_star_prime - logp_prime))

    return era_loss.mean()
```

## C  Details for molecular generator experiments

### C.1  Pretraining details

In this work, we represent all molecules as SMILES strings and tokenize SMILES strings according to the approach in Schwaller et al. [2018]. Our dataset consisted of all small-molecules from the ChEMBL database that were of length 500 tokens or less. Ultimately, this token limit filtered out approximately 0.1% of the small-molecules in the original ChEMBL dataset. The alphabet generated from this curated dataset consists of 324 tokens, which we augmented with start, stop, and padding tokens.

We first pretrained a model according to a next-token prediction, self-supervised learning approach. We trained a model using the standard cross entropy loss

$$\mathcal{L}_{\text{CE}} = -\sum_{t=1}^{T} \log p_\theta(\boldsymbol{x}_{t+1}|\boldsymbol{x}_{1:t}). \tag{33}$$

Our trained molecular generator consisted of just the encoder block of a standard multi-head attention transformer Vaswani et al. [2017]. Finally, the model had 2 layers, 8 heads, and a width of 512. For pretraining, we used an Adam optimizer with a learning rate of $1.0 * 10^{-5}$. We emphasize that this pretrained generator samples molecules in an unprompted fashion; given just a start-of-sequence token, we can autoregressively generate a sequence of tokens. Moreover, it is possible that this sequence of tokens corresponds to a molecule that is not chemically valid, and we find that around 88% of all generated molecules are chemically valid. Lastly, we measure the diversity of the pretrained molecular generator by first generating 1500 molecules and then computing the Tanimoto similarity between every pair of molecules. We plot the distribution of all pairwise Tanimoto similarities from

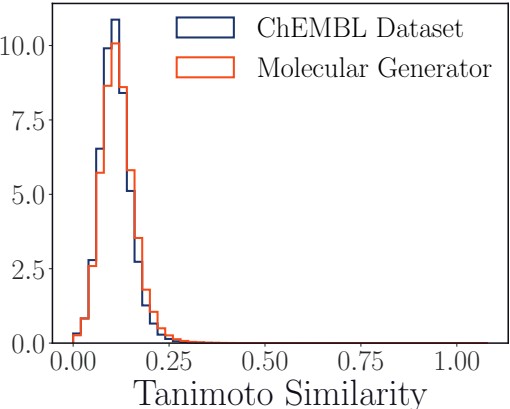

Figure S7: Chemical diversity of samples from training dataset and from unprompted molecular generator (unaligned) as measured by pairwise Tanimoto similarities. Lower Tanimoto similarities correspond to more chemically dissimilar molecules.

| Property name ($f$) | Energy function ($U$) |
|---|---|
| Tanimoto similarity | $U = -\log(f(\boldsymbol{y}))$ |
| QED | $U = -\log(f(\boldsymbol{y}))$ |
| Docking Oracles (GSK3$\beta$ and JNK3) | $U = -\log(f(\boldsymbol{y}))$ |
| Wildman-Crippen LogP | $U = (f(\boldsymbol{y}) - \mu)/2\sigma^2$ |
| Molar refractivity | $U = (f(\boldsymbol{y}) - \mu)/2\sigma^2$ |
| Ring count | $U = (f(\boldsymbol{y}) - \mu)/2\sigma^2$ |

Table S2: Definitions of energy functions (in reduced units) used for each of the five chemical properties investigated in this work. Here $\boldsymbol{y}$ refers to the generated molecule.

this sample and from all pariwise Tanimoto similarities from 1500 randomly sampled molecules from the original dataset in Fig. S7. We observe that we can generate molecules that are quite distinct (i.e. low Tanimoto similarity) in comparison with all other molecules.

## C.2 Chemical properties

We investigated aligning the molecule generator to several target chemical properties, which we detail below. All of the properties can be easily computed using either the RDKit package or the tdc Huang et al. [2021] package. We list the energy function and parameters used for the corresponding energy functions for each of these properties in Table S2.

Tanimoto similarity is a measure of chemical and structural properties between two molecules and ranges from 0 to 1, where higher values correspond to more similar molecules Rogers and Tanimoto [1960]. Quantitative estimation of drug-likeness (QED) is evaluated by taking the geometric mean of a set of "desirability functions" for different molecular descriptors and also ranges continuously from values of 0 to 1 Bickerton et al. [2012], where higher values correspond to more drug-like molecules. The octanol-water parition coefficient (Wildman-Crippen LogP) is a measure of hydrophobicity frequently employed in medicinal chemistry applications Wildman and Crippen [1999]. Molecules with more positive values are more hydrophobic (i.e. more soluble in octanol relative to water), whereas molecules with more negative values are more hydrophilic (i.e. more soluble in water relative to octanol). Molar refractivity is similarly calculated as a linear combination of atomic contributions, and is a positive number that serves as a measure for molecular size and polarizability Wildman and Crippen [1999]. A higher molar refractivity corresponds to larger and more polarizable molecules. Finally, ring count corresponds to the number of rings in a molecule.

Under the definitions of the energy functions in Table S2, it is possible for a generated sequence to not be chemically valid. For these cases, we manually define energies that are sufficiently high to penalize that outcome and we report these values in Table S3. Furthermore, when the computed

| Property name ($f$) | Energy |
|---|---|
| Tanimoto similarity | 10 |
| QED | 4.5 |
| Docking Oracles (GSK3$\beta$ and JNK3) | 4.5 |
| Wildman-Crippen LogP | 300 |
| Molar refractivity | 400 |
| Ring count | 70 |

Table S3: Property-specific energy values (in reduced units) used to treat chemically invalid sequences.

QED or Tanimoto Similarity is 0, the energy is infinite, and to ensure numerical stability, we set the value of the energies to be 4.5 and 10 respectively. Finally, in the prompted molecular generator experiments in Section 4.1.2, we assign an energy of 3.5 to the setting where Tanimoto similarity between the generated and prompt molecule is 1.0 (i.e they are the same) in order to penalize this outcome. Here, all energy and $\beta$ values are reported in reduced units.

## C.3 Molecular alignment details

### C.3.1 ERA ablations and comparison to DPO

| $\beta$ | $\gamma$ | Validity ↑ | Top 100 Mean ↑ | Top 100 Diversity ↓ |
|---|---|---|---|---|
| 10 | 0.01 | 72.00 | 0.932 | 0.129 |
| 10 | 0.1 | 74.23 | 0.933 | 0.131 |
| 100 | 0.01 | 74.71 | 0.934 | 0.129 |
| 100 | 0.1 | 75.14 | 0.935 | 0.131 |

Table S4: Descriptive statistics of 10,000 generated SMILES after aligning the molecular transformer model with ERA for QED maximization across several values of $\beta$ and $\gamma$.

| **DPO results** | | | |
|---|---|---|---|
| $\beta_{\text{DPO}}$ | Validity (%) ↑ | Top-100 mean ↑ | Top-100 diversity ↓ |
| 100 | 86.91 | 0.936 | 0.131 |
| 10 | 80.92 | 0.943 | 0.134 |
| 1 | 82.23 | 0.943 | 0.156 |
| 0.1 | 54.46 | 0.947 | 0.174 |
| **ERA results** ($\gamma = 9$) | | | |
| $\beta_{\text{ERA}}$ | Validity (%) ↑ | Top-100 mean ↑ | Top-100 diversity ↓ |
| 0.1 | 84.78 | 0.939 | 0.149 |
| 1 | 85.73 | 0.936 | 0.145 |
| 10 | 79.07 | 0.932 | 0.141 |
| 100 | 88.83 | 0.941 | 0.145 |

Table S5: Comparison of DPO and ERA results across different $\beta$ values on the task of QED maximization.

To evaluate the effects of the parameters $\beta$ and $\gamma$ on the performance of ERA, we assess the algorithm's performance on the task of QED maximization by ablating $\beta$ and $\gamma$ in Table S4. We observe that higher $\beta$ leads to greater dominance of the reward and increasing values for the Top-100 mean QED whereas higher $\gamma$ leads to higher validity due to the increased regularization against the reference policy, both consistent with the minimizer presented in 5.

To assess ERA against DPO, we evaluate each method on their performance on the task of QED maximization. We note that the DPO objective does not contain an explicit reward component and instead only maximizes the margins between preferred and dis-preferred samples. Additional care needs to be taken when comparing DPO and ERA hyperparameters as $\beta_{ERA} \neq \beta_{DPO}$ due to a

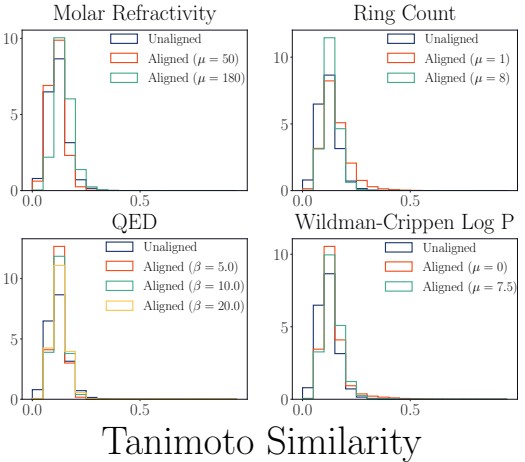

Figure S8: Chemical diversity of samples from unprompted molecular generator after alignment as measured by pairwise Tanimoto similarities. (See Fig. 2, Section 4.1.1)

difference in definition. Specifically, the minimizer of the DPO objective is:

$$\pi_{DPO}(x, y) \propto \exp(-\beta_{DPO}^{-1} r(x, y) + \log \pi_{ref})$$

Whereas for ERA, the minimizer is

$$\pi_{ERA}(x, y) \propto \exp\left( \frac{-\beta_{ERA}}{1 + \gamma} r(x, y) + \frac{\gamma}{1 + \gamma} \log \pi_{ref} \right)$$

The formulation of the DPO objective does not permit explicitly scaling the contribution of the regularization term against the prior while ERA permits this. This also implies that

$$\beta_{ERA} = \frac{1 + \gamma}{\beta_{DPO}}$$

Furthemore, we take $\gamma$ to be relatively large since as $\gamma \to \infty$, $\frac{\gamma}{1+\gamma} \to 1$. In our comparisons, we choose $\gamma = 9$. Table S5 contains descriptive statistics of 10,000 generated SMILES after aligning the molecular transformer model for QED maximization using either DPO or ERA across several matched values of $\beta$. It is arranged such that the corresponding rows can be compared directly against each other. We did not scan any $\beta < 0.1$ since the validity in those cases was low for both methods. Due to parameter matching, we expect ERA and DPO to be similar here: validity is comparable, with ERA surpassing DPO at higher $\beta_{ERA}$, as well as the top-100 average QED. ERA consistently generates diverse molecules across values of $\beta_{ERA}$ as opposed to DPO, as evidenced by the lower similarity among the top-100 generations for $\beta_{ERA} = 10$ and $\beta_{ERA} = 100$. This again reflects the fact that the entropy-regulated ERA objective promotes diverse generations while effectively avoiding greedy policies as opposed to DPO, which focuses only on maximizing the margins between preferred and dispreferred sequences. This highlights a key advantage of ERA since good molecular diversity is a key consideration for chemical and drug discovery tasks.

### C.3.2 Unprompted molecular generation (RDKit oracles)

We first investigated aligning the unprompted molecular generator to sample small-molecules with desired properties. We carried out alignment using the property-specific energies described in Table S2. All alignment properties were initialized with the weights of the pretrained model and trained using an Adam optimizer with learning rate $1.0 * 10^{-6}$. We tabulate the chemical validity for single-property alignment in Table S6 and for multi-property alignment in Table S7. While we do see a drop in chemical validity after alignment, we see that a majority of the samples we generate post-alignment are still chemically valid despite no regularization to a reference policy. We measure the chemical diversity for these experiments by computing all pairwise Tanimoto similarities from all chemically valid predictions of 1500 generated molecules. We visualize the chemical diversity for single-property experiments in Fig. S8 and multi-property experiments in Fig. S9. We observe that the samples are still highly diverse chemically after alignment. All plots in Fig. 2 and Fig. 3 were computed using 1500 generated molecules per experiment.

| Property name | Hyperparameters | Chemical validity |
|---|---|---|
| Unaligned | N/A | 88% |
| Molar Refractivity | $\beta = 1.0, \mu = 50, \sigma = 10, \gamma = 0.0$ | 82% |
| Molar Refractivity | $\beta = 1.0, \mu = 180, \sigma = 10, \gamma = 0.0$ | 74% |
| Ring Count | $\beta = 1.0, \mu = 1, \sigma = 1.0, \gamma = 0.0$ | 84% |
| Ring Count | $\beta = 1,0, \mu = 8, \sigma = 1.0, \gamma = 0.0$ | 59% |
| LogP | $\beta = 10.0, \mu = 2.5, \sigma = 1.0, \gamma = 0.0$ | 74% |
| LogP | $\beta = 10.0, \mu = 7.5, \sigma = 1.0, \gamma = 0.0$ | 63% |
| QED | $\beta = 5.0, \gamma = 0.0$ | 54% |
| QED | $\beta = 10.0, \gamma = 0.0$ | 66% |
| QED | $\beta = 20.0, \gamma = 0.0$ | 65% |

Table S6: Percentage of generated sequences that were chemically valid for samples from unprompted molecular generator after alignment. (See Fig. 2, Section 4.1.1).

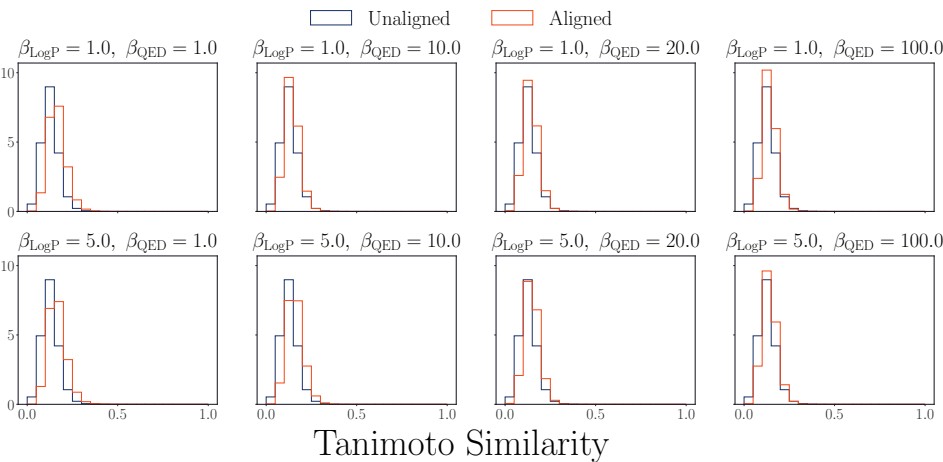

Figure S9: Chemical diversity of samples from unprompted molecular generator after multi-property alignment as measured by pairwise Tanimoto similarities. (See Fig. 3, Section 4.1.1).

| Hyperparameters | Chemical validity |
|---|---|
| Unaligned | 88% |
| $\beta_{\mathrm{QED}} = 1.0, \beta_{\mathrm{LogP}} = 1.0, \mu_{\mathrm{LogP}} = 7.5, \sigma_{\mathrm{LogP}} = 1.0, \gamma = 0.0$ | 60% |
| $\beta_{\mathrm{QED}} = 1.0, \beta_{\mathrm{LogP}} = 10.0, \mu_{\mathrm{LogP}} = 7.5, \sigma_{\mathrm{LogP}} = 1.0, \gamma = 0.0$ | 67% |
| $\beta_{\mathrm{QED}} = 1.0, \beta_{\mathrm{LogP}} = 20.0, \mu_{\mathrm{LogP}} = 7.5, \sigma_{\mathrm{LogP}} = 1.0, \gamma = 0.0$ | 68% |
| $\beta_{\mathrm{QED}} = 1.0, \beta_{\mathrm{LogP}} = 100.0, \mu_{\mathrm{LogP}} = 7.5, \sigma_{\mathrm{LogP}} = 1.0, \gamma = 0.0$ | 63% |
| $\beta_{\mathrm{QED}} = 5.0, \beta_{\mathrm{LogP}} = 1.0, \mu_{\mathrm{LogP}} = 7.5, \sigma_{\mathrm{LogP}} = 1.0, \gamma = 0.0$ | 64% |
| $\beta_{\mathrm{QED}} = 5.0, \beta_{\mathrm{LogP}} = 10.0, \mu_{\mathrm{LogP}} = 7.5, \sigma_{\mathrm{LogP}} = 1.0, \gamma = 0.0$ | 62% |
| $\beta_{\mathrm{QED}} = 5.0, \beta_{\mathrm{LogP}} = 20.0, \mu_{\mathrm{LogP}} = 7.5, \sigma_{\mathrm{LogP}} = 1.0, \gamma = 0.0$ | 62% |
| $\beta_{\mathrm{QED}} = 5.0, \beta_{\mathrm{LogP}} = 100.0, \mu_{\mathrm{LogP}} = 7.5, \sigma_{\mathrm{LogP}} = 1.0, \gamma = 0.0$ | 68% |

Table S7: Percentage of generated sequences that were chemically valid for samples from unprompted molecular generator after multi-property alignment. (See Fig. 3, Section 4.1.1).

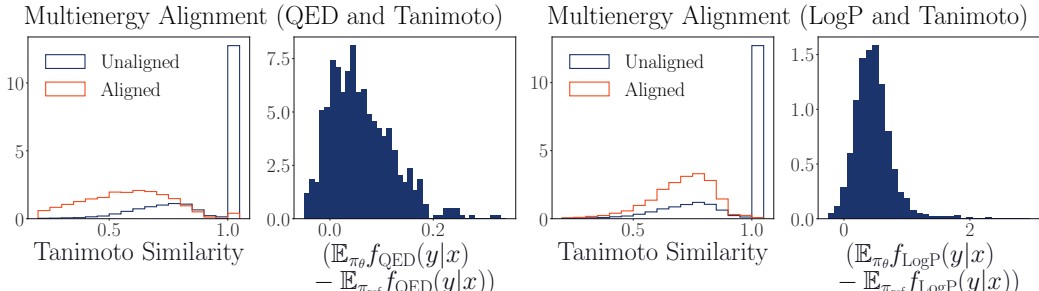

Figure S10: Prompted multi-property molecular generator alignment. From left to right: Tanimoto similarities computed between the prompt and sampled molecules for both aligned and unaligned policies (QED and Tanimoto alignment), per-prompt difference in the average QED under aligned and unaligned policies (QED and Tanimoto alignment), Tanimoto similarities computed between the prompt and sampled molecules for both aligned and unaligned policies (LogP and Tanimoto alignment), and per-prompt difference in the average LogP under aligned and unaligned policies (LogP and Tanimoto alignment). With alignment, we target higher QED and LogP values, while still sampling molecules chemically similar—but not identical to—the prompt molecule.

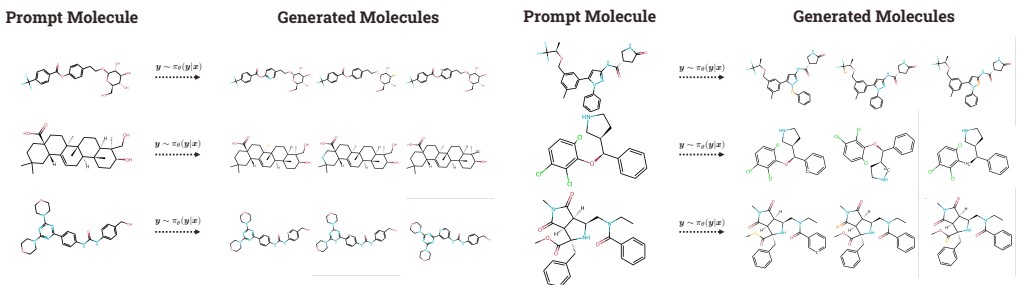

Figure S11: Sample molecules from prompted molecular generator after multi-property alignment experiments: QED and Tanimoto (left) and LogP and Tanimoto (right). With alignment, generated molecules are diverse, while still chemically similar to prompt molecule.

### C.3.3 Prompted molecular generation (RDKit oracles)

Next, we generate small-molecules with desired properties conditioned on a prompt, where the prompt is itself another molecule. In the experiments here, we consider the setting where we generate molecules that are chemically similar to the prompt molecule. With this in mind, we first carry out a fine-tuning step using a synthetic dataset $\mathcal{D} = \{(\boldsymbol{x}_1, \boldsymbol{y}_1), \ldots, (\boldsymbol{x}_n, \boldsymbol{y}_n)\}_{i=1}^{N}$, where $\boldsymbol{x}$ corresponds to the SMILES string of a prompt molecule and $\boldsymbol{y}$ corresponds to the SMILES string of the conditionally generated molecule. To curate this dataset, we consider all molecules in our original filtered ChEMBL dataset to be a prompt molecules and for each prompt molecule $\boldsymbol{x}_i$, we generate a response molecule $\boldsymbol{y}_i$ by simply perturbing a random token from $\boldsymbol{x}_i$. If the perturbed sequence was chemically invalid, we repeated the random perturbation until a valid molecule was generated. The prompted generator was the same size as the unprompted molecular generator, and we initialized the weights using those of the pre-trained unprompted molecular generator. We then carried out supervised fine-tuning using an Adam optimizer with learning rate $1.0 * 10^{-5}$ and used this generator as our reference policy for all prompted alignment experiments. All plots in Fig. S10 were computed using 100 generated

| Hyperparameters | Chemical validity |
|---|---|
| Unaligned | 93% |
| $\beta_{\text{Tanimoto}} = 5.0, \beta_{\text{LogP}} = 10.0, \mu_{\text{LogP}} = 5.0, \sigma_{\text{LogP}} = 1.0, \gamma = 0.1$ | 91% |
| $\beta_{\text{Tanimoto}} = 5.0, \beta_{\text{QED}} = 500.0, \gamma = 0.1$ | 81% |

Table S8: Percentage of generated sequences that were chemically valid for samples from prompted molecular generator after multi-property alignment. (See Fig. S10, Section 4.1.2).

molecules per prompt, where we carried inference over 500 prompts per experiment. We tabulate the chemical validity of the prompted generator in Table S8.

### C.3.4 Unprompted molecular generation (protein-ligand docking oracles)

| | GSK3$\beta$ top-10 AUC | JNK3 top-10 AUC |
|---|---|---|
| **ERA** | **0.985 ± 0.001** | **0.989 ± 0.002** |
| REINVENT | 0.865 ± 0.043 | 0.783 ± 0.023 |
| GraphGA | 0.788 ± 0.070 | 0.553 ± 0.136 |
| PPO | 0.90 ± 0.02 | 0.80 ± 0.04 |
| PPOD | 0.92 ± 0.02 | 0.87 ± 0.02 |

Table S9: Top-10 AUC scores on GSK3$\beta$ and JNK3 tasks averaged across 5 random seeds. Compared to state-of-the-art methods as reported in Gao et al. [2022] and Bou et al. [2024], ERA has higher sampling efficiency. Results for compared methods are reproduced from Gao et al. [2022] and Bou et al. [2024]

For the work here, we used a computational oracle that predicts the docking score for two kinases, JNK3 and GSK3$\beta$, where these oracles were defined using the `tdc` package. We first carried out a supervised fine-tuning step using all molecules in ChemBL with an oracle score above 0.5. For the JNK3 model, we carried out fine-tuning on a dataset of size 7386 molecules, and for $GSK3\beta$, we carried out fine-tuning on a dataset of size 43381 using the Adam optimizer with a learning rate of $1.0 * 10^{-5}$. From these fine-tuned models, we sampled 40000 molecules, evaluated the oracle scores, and used this dataset to carry out alignment using all possible pairs of molecules. For any invalid molecule, we assign an energy of 4.5. All alignment runs were done using the Adam optimizer with a learning rate of $1.0 * 10^{-6}$. We observed a sampled validity of 74% on the model aligned for GSK3$\beta$ and a sample validity of 93.6% for the model aligned for JNK3.

We compute metrics on the top-100 novel and unique molecules on 20000 sampled molecules (see Figure S12) from the aligned models. We compute the mean score and the internal diversity score (IntDiv) Hu et al. [2023] computed according to the following

$$\text{IntDiv}(A) := \frac{1}{|A|(|A|-1)} \sum_{(x,y) \in A \times A, x \neq y} T(x,y), \tag{34}$$

where $A$ is a set of compounds and $T$ represents the Tanimoto similarity. Lower IntDiv scores correspond to more diverse molecules.

We additionally compute sample efficiency using the top-10 AUC metric (see Table S9), which is the area under the curve (AUC) of the mean property value of the top-10 performing molecules versus the number of oracle calls (see Figure S13). We note that once we reach a top-10 average of 1.0, we do not make further oracle calls as subsequent oracle calls will not change the top-10 average and will artificially inflate the AUC.

### C.3.5 Analysis of invalid generations

Especially when the regularization $\gamma = 0$, we see that ERA reduces validity of the generated SMILES. There are some limitations imposed by the fact that most of our downstream characterizations cannot be run on invalid SMILES string because of the failure of RDKit to parse. To assess the failure modes of these invalid generations, we computed two measures of the diversity: an estimate of the Levenshtein distance and an estimate of the Shannon entropy for both valid and invalid molecules.

Table S10 demonstrates properties of the valid and invalid generations from a model aligned using ERA for QED maximization for $\beta = 100.0$, $\gamma = 0.1$. Notably, the Shannon entropy of generated tokens and the mean Levenshtein distance between all of the generated SMILES is higher for the invalid outputs than the valid ones. These metrics suggest that there is in fact less clustering around one specific region of chemical space in these failure cases.

For the same QED maximization run ($\beta = 100.0$, $\gamma = 0.1$) as above, we found a significant failure mode to be that of missing tokens to close branches (67.6% of all invalid generations). Furthermore, failure to properly close rings also occurs in a signifcant amount of invalid generations (30.5%). One potential failure mode that seems to be relatively insignificant is that of failing to predict the stop token

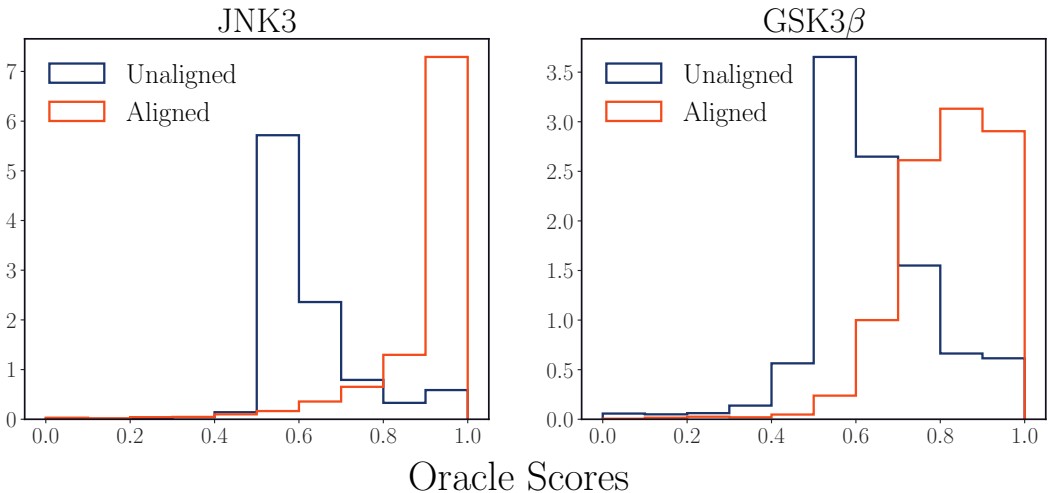

Figure S12: Distribution of GSK3$\beta$ and JNK3 oracle scores sampled from unaligned reference model and aligned model ($\beta = 100.0$, $\gamma = 0.0$). 20k molecules were sampled from each model and only oracle scores of valid molecules are plotted.

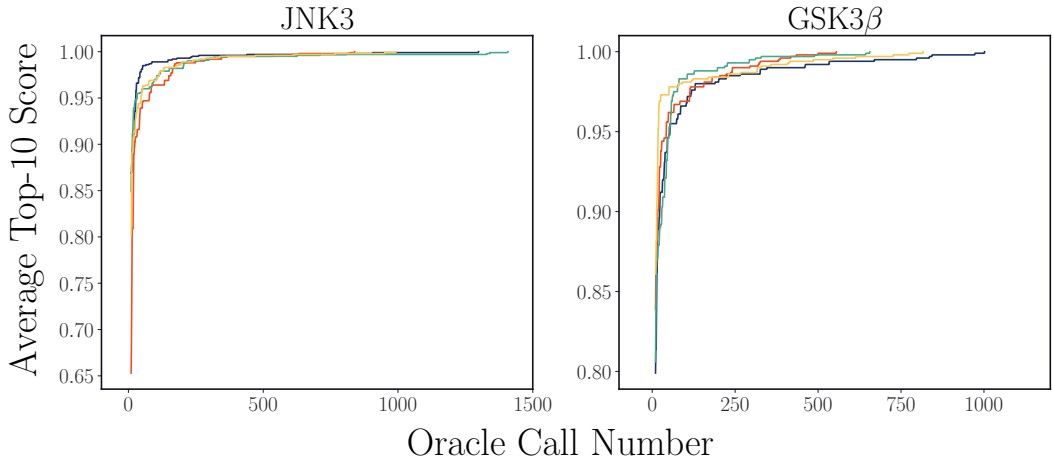

Figure S13: The average score of top-10 performing *valid, novel*, and *unique* molecules as a function of the number of oracle calls made to the aligned models. Scores are computed using the JNK3 and GSK3$\beta$ oracles, respectively, for five different random seeds. Samples that are invalid, present in the dataset, or already previously sampled are discarded and do not count towards an oracle call.

before our hard limit of 500 tokens, which occurs in fewer than 1% of invalid generations. Beyond this analysis, visual inspection of several failure cases does not appear to reveal any particular motif that is repeated among these examples beyond the model attempting to generate highly branched molecules or molecules with several rings.

| Generations | Shannon Entropy | Levenshtein Distance |
|---|---|---|
| Valid | 2.2574 | 39.8225 |
| Invalid | 2.3177 | 56.1066 |

Table S10: Shannon entropy and Levenshtein distances for valid and invalid generations from a QED maximization molecular generator alignment experiment with $\beta = 100.0$, $\gamma = 0.1$.

# D Details for protein language model experiments

We consider mutating the TrpB protein at 4 sites (positions 182, 183, 184, and 186) to all of the 20 standard amino acids and compute the EVMutation score Hopf et al. [2017] for all $20^4 = 160000$ sequences (see Yang et al. [2023] for dataset). We used the standard tokenization scheme in ESM3 and during training and inference masked all non-sequence tasks Hayes et al. [2024]. ESM3 is a bidirectional transformer and so to generate mutant sequences, we first pass in the full sequence with the mutant sites masked and use the model to "unmask" these four sites simultaneously. For the sequence track, ESM3 also contains additional tokens representing non-standard residues and special cases, and we consider sequences "invalid" if the generated token does not correspond to one of the 20 standard amino acid sequences.

When sampling mutant sequences, the native ESM3-1.4B model does not generate a diverse set of sequences and so we first synthesize an initial dataset of 512 random mutant sequences as would normally be done in a random mutagenesis experiment. We did not carry out any supervised fine-tuning step and here considered the energies to be the negative of the EVMutation score and evaluated log probabilities on the pretrained ESM3-1.4B model. We then carried alignment using ERA, where our dataset consisted of all possible unique pairs of these 512 sequences. For the experiments here, we used the RMSProp optimizer with a learning rate of $1.0 * 10^{-5}$. We plot the EVMutation score of 512 generated sequences across various $\beta$ values and $\gamma = 0.001$ in Figure 5. Finally, we note that we did not sample any "invalid" sequences as defined above.

# E Computational resources

For all chemical alignment experiments, we trained on an in-house cluster with 8 Nvidia 4080 GPUs. For ESM3 experiments, we used resources of the National Energy Research Scientific Computing Center (NERSC), a Department of Energy Office of Science User Facility. Jobs run on NERSC used at most 4 Nvidia A100 GPUs (either 40GB or 80GB depending on what was allocated).

# F Societal and broader impacts

The ERA algorithm we have introduced in this work is a powerful and scalable approach towards generating outputs targeting some desired combination of properties. In this work we have demonstrated the efficacy of this method in both a chemical context and a language context. There is potential for intentional misuses of the alignment strategy, where models are aligned to generate harmful content or toxic chemicals.

