# OpenReview forum: "Aligning Transformers with Continuous Feedback via Energy Rank Alignment"
_NeurIPS.cc/2025/Conference — NeurIPS 2025 poster_

### Official Review · Reviewer_EGLr · 2025-07-03

**Clarity:** 2
**Significance:** 3
**Originality:** 3
**Rating:** 5
**Confidence:** 3

**Summary:**

In materials and drug discovery, we want to find molecules that maximize some desired properties, such as docking score and solubility. Such processes amount to searching the vast chemical space of molecules.


This paper draws a similarity between search in chemical space and language model (LM) alignment. Indeed, alignment in LMs amounts to “biasing” a search policy towards sequences with high rewards or desired properties. Specifically, the authors propose the energy rank alignment (ERA) algorithm that leverages an explicit reward function that can be used to optimize autoregressive policies in the chemical space.


ERA draws similarity to standard reinforcement learning from human feedback algorithms such as PPO and DPO. Moreover, it is shown to be effective in searching through diverse parts of chemical space.

**Questions:**

Can the authors elaborate on the “design decisions” of the algorithm in a more compelling way? (See “Weaknesses” above for context.)

**Ethical Concerns:**

["NO or VERY MINOR ethics concerns only"]

**Final Justification:**

- The paper’s significance and novelty are good
- Clarity issues have been addressed well during the rebuttal
- The authors have also clarified their algorithm’s costs
- After seeing other reviews and the authors’ responses, I don’t see any significant issues anymore

**Limitations:**

If the costs are a limitation of the algorithm, I suggest that the authors add a discussion in this section.

**Quality:**

2

**Strengths And Weaknesses:**

**Strengths**


1. The application of LM alignment in molecule generation is very interesting.


2. The proposed method seems to be effective for aligning molecular LM.


**Weaknesses**


1. I think the main issue with the paper is its presentation (Sec. 2). All the equations are simply stated, without much step-by-step derivation. This makes it hard to understand the motivation behind the different quantities presented. E.g. in Eq. 2, the authors simply stated “To impose the preferences, we minimize the objective: …” — it would be much better if the authors first derive/motivate each term (e.g. why the integrals?) in Eq. 2. Note that, this is not just about Eq. 2; I highly encourage the authors to revisit Sec. 2 as a whole.


2. Connected to the point above, the discussion in Sec. 2 lacks theoretical justification/analysis. It would be great if the authors can show some theoretical results there. I know that some results might have been deferred to the appendix, but I encourage the authors to state some of them in the main text.


3. Presentation of the results can also be significantly improved. E.g., it is quite hard to understand what Fig. 2 is trying to communicate, esp. since it is quite far away from Sec. 3.2, where it is being discussed.


4. Last but definitely not least, costs are very important in materials/drug discovery. The authors need to discuss them. For example, through a wall-clock runtime vs. performance plot, and show that the proposed algorithm is on the Pareto frontier.

---

> ### Author Rebuttal · Authors · 2025-07-31
>
> > I think the main issue with the paper is its presentation (Sec. 2). All the equations are simply stated, without much step-by-step derivation. This makes it hard to understand the motivation behind the different quantities presented. E.g. in Eq. 2, the authors simply stated “To impose the preferences, we minimize the objective: …” — it would be much better if the authors first derive/motivate each term (e.g. why the integrals?) in Eq. 2. Note that, this is not just about Eq. 2; I highly encourage the authors to revisit Sec. 2 as a whole.
>
> > Can the authors elaborate on the “design decisions” of the algorithm in a more compelling way? (See “Weaknesses” above for context.)
>
> We agree that the derivation and motivation of our approach would benefit from a more thorough explanation in the main text. If accepted, we will expand Section 2 so that it includes a systematic derivation of the approach and clear articulation of the advantages over alternative approaches.
>
> Briefly, the main difference in the goal of ERA is to leverage an explicit reward function to balance sample diversity and "greedy" optimization to maximize rewards. In both PPO and DPO, the prior term cannot be independently tuned at fixed temperature. More mathematically, the minimizer of the DPO objective is,
> $$
> \pi_{\textrm{DPO}}(x,y) \propto \exp \left( \beta_{\rm DPO}^{-1} r(x,y) + \log \pi_{\rm ref} \right),
> $$
> whereas the minimizer of the ERA objective is
> $$
> \pi_{\textrm{ERA}}(x,y) \propto \exp \left( \frac{\beta_{\rm ERA}}{1+\gamma} r(x,y) + \frac{\gamma}{1+\gamma}\log \pi_{\rm ref} \right).
> $$
> In DPO, the magnitude of the regularization to the prior cannot be tuned independently of $\beta$. In particular, this means that the only way to change the impact of the prior is to effectively lower the sampling temperature and hence lower sample diversity.
> On the other hand ERA allows these parameters to be tuned independently, for example, $\gamma\to 0$ enables a distribution with no prior regularization while preserving sample diversity.
>
> The inclusion of an explicit reward model is the second important aspect of ERA. The explicit reward matters most when the number of observations of pairwise comparisons is small. **Stated simply, DPO overly penalizes molecules with similar reward values by maximizing the margin between them.**
>
> For example, if the comparison of two unique molecules shows that $\boldsymbol{y} \succ \boldsymbol{y}'$, i.e., $\boldsymbol{y}$ is preferred to $\boldsymbol{y}'$, then the DPO requires that the optimal policy requires that
> \begin{equation}
>     \pi_{\star}^{\textrm{DPO}}(\boldsymbol{y}|\boldsymbol{x}) = 1 \quad \textrm{and} \quad \pi_{\star}^{\textrm{DPO}}(\boldsymbol{y}'|\boldsymbol{x}) = 0.
> \end{equation}
> **This is true even if both molecules have similar, high reward values.**
>
> This can also lead to instability: the sampled gradients of the objective used for DPO are proportional to the implicit reward discrepancy,
> \begin{equation}
>     \nabla_{\boldsymbol{\theta}} \hat{\mathcal{L}}^{\textrm{DPO}}(\boldsymbol{y}, \boldsymbol{y}', \boldsymbol{x}) = \sigma \left( \beta^{-1}\gamma \left[\log \frac{\pi(\boldsymbol{y}'|\boldsymbol{x})}{\pi_{\rm ref}(\boldsymbol{y}'|\boldsymbol{x})} - \log \frac{\pi(\boldsymbol{y}|\boldsymbol{x})}{\pi_{\rm ref}(\boldsymbol{y}|\boldsymbol{x})} \right] \right) \nabla_{\boldsymbol{\theta}} \log \frac{\pi(\boldsymbol{y}|\boldsymbol{x})}{\pi(\boldsymbol{y}'|\boldsymbol{x})},
> \end{equation}
> which when $\pi(\boldsymbol{y}'|\boldsymbol{x}) \to 0$, leads to instability as $-\log \pi(\boldsymbol{y}'|\boldsymbol{x}) \to \infty.$ On the other hand, the ERA gradients are scaled by the relative preference discrepancy,
> \begin{equation}
>     \nabla_{\boldsymbol{\theta}} \mathcal{L}^{\textrm{ERA}}(\boldsymbol{y}, \boldsymbol{y}', \boldsymbol{x}) = \left( \frac{1-\sigma_{\star}(\boldsymbol{y}\succ\boldsymbol{y}'|\boldsymbol{x})}{1-\sigma_{\boldsymbol{\theta}}(\boldsymbol{y}\succ\boldsymbol{y}'|\boldsymbol{x})} - \frac{\sigma_{\star}(\boldsymbol{y}\succ\boldsymbol{y}'|\boldsymbol{x})}{\sigma_{\boldsymbol{\theta}}(\boldsymbol{y}\succ\boldsymbol{y}'|\boldsymbol{x})} \right) \nabla_{\boldsymbol{\theta}} \sigma_{\boldsymbol{\theta}}(\boldsymbol{y}\succ\boldsymbol{y}'|\boldsymbol{x}).
> \end{equation}
> The advantage of a reward model becomes apparent because
> \begin{equation}
>     \sigma_{\star}(\boldsymbol{y}\succ\boldsymbol{y}'|\boldsymbol{x}) = p_{\gamma}(\boldsymbol{y}\succ \boldsymbol{y}'|\boldsymbol{x}) = \sigma \left(  \frac{\beta}{1+\gamma} \left[ (U(\boldsymbol{x},\boldsymbol{y}') - U(\boldsymbol{x},\boldsymbol{y})) +  \beta^{-1}\gamma \log \frac{\pi_{\rm ref}(\boldsymbol{y}|\boldsymbol{x})}{\pi_{\rm ref}(\boldsymbol{y}'|\boldsymbol{x})} \right] \right)
> \end{equation}
> and hence the optimum of $\mathcal{L}^{\textrm{ERA}}$ will not lead to policies in which high reward samples are suppressed by the model. This could alternatively be implemented by taking $\beta\to\infty$. Choosing an appropriate reward model, hence, gives the flexibility to control instability if it becomes problematic.
>
>
> > Costs are very important in materials/drug discovery. The authors need to discuss them. For example, through a wall-clock runtime vs. performance plot, and show that the proposed algorithm is on the Pareto frontier.
>
>
> Indeed, costs are extremely important in this context. In our submission, we discuss two types of costs: first the computational costs associated with inference for molecular generation, which are low:
> > Line 237: The inference costs are notably low for our approach; sampling 20000 molecules and filtering takes only minutes on a single GPU.
>
> The more relevant costs for molecular optimizations are typically the number of oracle calls, which might be experimentally or computationally expensive to evaluate. We quantified this in our submission (Fig. S13) by looking at top-10 statistics as a function of number of oracle calls. The following Table (original submission Table S7) demonstrates that the sample efficiency of ERA is superior to REINVENT and GraphGA.
>
>
> |          | GSK3β top-10 AUC | JNK3 top-10 AUC |
> |----------|------------------|------------------|
> | **ERA**  | **0.985 ± 0.001** | **0.989 ± 0.002** |
> | REINVENT | 0.865 ± 0.043 | 0.783 ± 0.023 |
> | GraphGA  | 0.788 ± 0.070 | 0.553 ± 0.136 |
>
>
>
> In our submission, we discussed this Table S7 and Figure S13 in lines 240-248.
> The top-10 AUC has previously been proposed as a key metric of performance relative to cost and we believe that our experiments indicate that ERA is state-of-the-art on this metric.

---

> > ### Comment · Reviewer_EGLr · 2025-08-08
> >
> > Thank you for the clarification. I highly suggest putting this step-by-step rationale/motivation for ERA in the next revision. The AUC table/figure should also be provided in the main text. They do make the proposed method much clearer and compelling!

---

> > > ### Author Response · Authors · 2025-08-08
> > >
> > > Thanks for your helpful comments, we agree and plan to include the explanation and the AUC in the main body of the paper!

---

### Official Review · Reviewer_7xzk · 2025-07-03

**Clarity:** 4
**Significance:** 3
**Originality:** 3
**Rating:** 3
**Confidence:** 3

**Summary:**

The paper introduces an algorithm called Energy Rank Alignment to adapt general molecular generative models to specific design objectives. The method formulates a direct optimization objective as an energy function derived from task-specific reward functions, and theoretically proves its convergence to the ideal Gibbs–Boltzmann distribution. Experimental results demonstrate the effectiveness of the proposed strategy across diverse scenarios, including both small molecule generation and protein sequence design.

**Questions:**

1. Figure 1 is not referenced in the main text. It is recommended that the authors cite and briefly discuss it to clarify its relevance.

2. The authors are encouraged to provide more intuition and explanations for each term in the key equations, particularly Equations (2)–(5), to improve readability and accessibility.

3. The font size used in the figures is relatively big and may hinder readability. It is suggested that the authors adjust the font size for better clarity.

**Ethical Concerns:**

["NO or VERY MINOR ethics concerns only"]

**Final Justification:**

I maintain my recommendation for rejection, as the rebuttal did not adequately address my concerns.

**Limitations:**

Yes.

**Paper Formatting Concerns:**

No.

**Quality:**

2

**Strengths And Weaknesses:**

The paper is well-written, featuring an intuitive introduction and detailed methodological description. The proposed approach presents a degree of novelty and is supported by experiments that demonstrate its effectiveness in guiding molecular generation toward desired objectives.

However, a major concern is the lack of comparison with other established post-adaptation strategies, such as fine-tuning, Direct Preference Optimization (DPO), and Proximal Policy Optimization (PPO). While the authors show that their method can effectively optimize generation outputs based on task-specific rewards, they do not provide empirical evidence of its advantages over these alternative techniques. Including such comparisons would significantly strengthen the claim of superiority and clarify the practical benefits of the proposed method.

In addition, the inclusion of quantitative metrics, rather than relying solely on visual figures, would provide a clearer assessment of the model’s performance. Further analysis and ablation studies of the proposed method would also be valuable to better understand its behavior and effectiveness.

---

> ### Author Rebuttal · Authors · 2025-07-31
>
> > However, a major concern is the lack of comparison with other established post-adaptation strategies, such as fine-tuning, Direct Preference Optimization (DPO), and Proximal Policy Optimization (PPO). While the authors show that their method can effectively optimize generation outputs based on task-specific rewards, they do not provide empirical evidence of its advantages over these alternative techniques. Including such comparisons would significantly strengthen the claim of superiority and clarify the practical benefits of the proposed method.
>
>
> For DPO, we note that the DPO objective does not contain an explicit reward component and instead only maximizes the margins between preferred and dis-preferred samples. Additional care needs to be taken when comparing DPO and ERA hyperparameters because $\beta_{DPO} \neq \beta_{ERA}$ due to a difference in definition. Specifically, the minimizer of the DPO objective is
> $$
> \pi_{DPO}(x, y) \propto \exp(-\beta_{DPO}^{-1}r(x, y) + \log \pi_{ref})
> $$
> Whereas for ERA, the minimizer is
> $$
> \pi_{ERA}(x, y) \propto \exp\left(\frac{-\beta_{ERA}}{1 + \gamma}r(x, y) + \frac{\gamma}{1 + \gamma}\log \pi_{ref}\right)
> $$
> The formulation of the DPO objective does not permit explicitly scaling the contribution of the regularization term against the prior while ERA permits this. This also implies that
> $$
> \beta_{ERA} = \frac{1 + \gamma}{\beta_{DPO}}
> $$
> Furthemore, we take $\gamma$ to be relatively large since as $\gamma \to \infty$, $\frac{\gamma}{1+\gamma} \to 1$. In our comparisons, we choose $\gamma = 9$. The tables below are arranged such that the corresponding rows can be compared directly against each other. We did not scan any $\beta < 0.1$ since the validity in those cases was low for both methods:
>
>
> **DPO results**
> | $\beta_{DPO}$ | Validity (%) $\uparrow$ | Top-100 mean $\uparrow$ | Top-100 diversity $\downarrow$ |
> | -------- | -------- | -------- | -------- |
> | 100      | 86.91     | 0.936     | 0.131
> | 10       | 80.92     | 0.943     | 0.134
> | 1        | 82.23     | 0.943     | 0.156
> | 0.1      | 54.46     | 0.947     | 0.174
>
> **ERA results ($\gamma = 9$)**
> | $\beta_{ERA}$ | Validity (%) $\uparrow$ | Top-100 mean $\uparrow$ | Top-100 diversity $\downarrow$ |
> |---------------|--------------------------|--------------------------|---------------------------------|
> | 0.1           | 84.78                    | 0.939                    | 0.149                           |
> | 1             | 85.73                    | 0.936                    | 0.145                           |
> | 10            | 79.07                    | 0.932                    | 0.141                           |
> | 100           | 88.83                    | 0.941                    | 0.145                           |
>
>
>
> The tables above contain descriptive statistics of 10,000 generated SMILES after aligning the molecular transformer model for QED maximization using either DPO or ERA across several matched values of $\beta$. We note that the validity in generations is comparable, with ERA maintaining consistently high generation validity and surpassing DPO at higher $\beta_{ERA}$, as well as the average QED among the top-100 generations. ERA also more consistently generates diverse molecules across values of $\beta_{ERA}$ as opposed to DPO as evidenced by the lower similarity among the top-100 generations for $\beta_{ERA} = 10$ and $\beta_{ERA} = 100$. This again reflects the fact that the entropy-regulated ERA objective promotes diverse generations while effectively avoiding greedy policies as opposed to DPO, which focuses only on maximizing the margins between preferred and dispreferred sequences. This highlights a key advantage of ERA since good molecular diversity is a key consideration for chemical and drug discovery tasks.
>
> > In addition, the inclusion of quantitative metrics, rather than relying solely on visual figures, would provide a clearer assessment of the model’s performance. Further analysis and ablation studies of the proposed method would also be valuable to better understand its behavior and effectiveness.
>
> Our submission contains several quantifications of the superiority of ERA compared to reinforcement learning strategies used in the literature. In our benchmarks, ERA outperforms RL-based strategies, including REINVENT and MolRL-mGPT, on the task of generating ligands for protein docking against GSK3$\beta$ and JNK3. These are techniques that use an objective similar to the one used in PPO. For completeness, we reproduce Table 1 of our submission here. The best numbers are bolded:
>
> | Method     | GSK3$\beta$ mean score $\uparrow$ | GSK3$\beta$ IntDiv $\downarrow$ | JNK3 mean score $\uparrow$ | JNK3 IntDiv $\downarrow$ |
> |:---------- | --------------------------------- |:-------------------------------:|:--------------------------:|:------------------------:|
> | ERA        | 0.996 $\pm$ 0.000                 |        **0.219 $\pm$ 0.002**        |     **0.987 $\pm$ 0.001**      |    **0.264 $\pm$ 0.005**     |
> | MolRL-MGPT | **1.000 $\pm$ 0.000**             |        0.362 $\pm$ 0.015        |     0.961 $\pm$ 0.010      |    0.372 $\pm$ 0.025     |
> | GFlowNet   | 0.649 $\pm$ 0.072                 |        0.715 $\pm$ 0.104        |     0.437 $\pm$ 0.219      |    0.716 $\pm$ 0.145     |
> | GraphGA    | 0.919 $\pm$ 0.016                 |        0.365 $\pm$ 0.024        |     0.875 $\pm$ 0.025      |    0.380 $\pm$ 0.015     |
> | JT-VAE     | 0.235 $\pm$ 0.083                 |        0.770 $\pm$ 0.067        |     0.159 $\pm$ 0.040      |    0.781 $\pm$ 0.127     |
> | REINVENT   | 0.965 $\pm$ 0.011                 |        0.308 $\pm$ 0.035        |     0.942 $\pm$ 0.019      |    0.368 $\pm$ 0.021     |
>
>
> Furthermore, we have added explicit comparisons to DPO as shown above, emphasized our previous comparisons to state of the art RL strategies for molecular optimization, and added additional parameter ablations to study the role of the inverse temperature parameter $\beta$ and the prior coupling strength $\gamma$. We emphasize that DPO constructs the reward function implicitly by design and we view ERA as the natural extension of DPO to the setting in which an explicit reward function is available.

---

### Official Review · Reviewer_zXeq · 2025-07-03

**Clarity:** 3
**Significance:** 3
**Originality:** 3
**Rating:** 5
**Confidence:** 4

**Summary:**

This paper introduces Energy Rank Alignment (ERA), a simple and effective method to fine-tune generative models so that they produce molecules or protein sequences with better properties, like higher drug-likeness or stronger binding affinity. Instead of using complex reinforcement learning or relying on human-labeled preferences, ERA takes advantage of a known reward function (e.g., QED, docking score) to decide which outputs are better. It compares pairs of samples generated by a pretrained model and teaches a new model to prefer the higher-scoring ones.

A key strength of ERA is that it works entirely offline — it doesn’t need to generate new data during training, making it much more efficient and practical for real-world use. The authors apply ERA to several tasks: molecule generation without prompts, lead optimization conditioned on a given molecule, docking-based bioactivity optimization, and protein sequence design. Across all tasks, ERA improves the quality of generated samples while keeping a good balance between validity and diversity.

In short, ERA offers a general-purpose, easy-to-apply method for aligning generative models with specific goals in molecular and protein design, backed by both theory and strong experimental results.

**Questions:**

1. **Sampling randomness during training**: The ERA framework relies on sampling multiple outputs \( y, y' \) from the reference model \( \pi_{\text{ref}}(y|x) \) to construct preference pairs. However, the paper does not specify the sampling strategy (e.g., temperature, top-k, top-p) used during this process. Could the authors clarify what sampling method was used, whether temperature was tuned, and whether different sampling settings significantly affect performance, diversity, or training stability?

2. **Validity drop after alignment**: In several tasks, the validity of generated SMILES strings drops after applying ERA (e.g., LogP↑ task). Could the authors provide more insight into this phenomenon? Specifically, do invalid generations tend to cluster around certain reward-optimized motifs or SMILES patterns? Is there evidence that the model is being pushed toward syntactically fragile or chemically implausible regions of molecular space?

3. **Analysis of failure cases**: It would be helpful to understand the nature of invalid or low-reward generations. Could the authors provide examples or visualizations of failure cases (e.g., invalid SMILES, chemically unstable structures) and discuss what types of prompts or optimization targets tend to induce them?

4. **Ablation of β and γ hyperparameters**: The ERA algorithm introduces two key hyperparameters — β for entropy shaping and γ for reference policy regularization — yet there is limited analysis of their effects. Could the authors provide a more detailed ablation study or discussion showing how varying these parameters affects the tradeoffs between reward, validity, and diversity across tasks? Additionally, is there any task-specific guidance on selecting appropriate values?

5. **Potential for constrained decoding**: Given that validity tends to decline slightly under strong reward optimization, have the authors considered integrating grammar-based decoding, constrained decoding, or validity filters during generation? If so, what tradeoffs were observed? If not, could this be a promising direction to mitigate invalid generations while preserving ERA’s reward-guided behavior?

**Ethical Concerns:**

["NO or VERY MINOR ethics concerns only"]

**Final Justification:**

I appreciate the authors’ solid rebuttal addressing all reviewers’ comments. I am happy to keep my original score.

**Limitations:**

- **Lack of ablation on key hyperparameters (β, γ):** The method introduces two central hyperparameters — β for controlling entropy (sharpness of reward preference) and γ for regularization toward the reference policy — but does not offer a detailed ablation or sensitivity analysis. Understanding how these values affect reward optimization, validity, and diversity is critical for reproducibility and for guiding practitioners applying ERA to new tasks.

- **Drop in chemical validity post-alignment not analyzed:** While validity is reported throughout, it often decreases after ERA alignment, especially in tasks like LogP optimization. The paper does not investigate this effect or analyze whether ERA leads the model to explore syntactically fragile or chemically unstable regions of molecule space.

- **Sampling randomness during training is under-specified:** Since ERA relies on comparing multiple samples \( y, y' \) drawn from the reference model for a given input \( x \), the quality and diversity of these samples depend heavily on the sampling procedure. However, the paper does not specify whether temperature, top-k, or nucleus sampling was used, nor does it analyze the impact of sampling strategy on training quality.

- **No demonstration of on-policy or adaptive extension:** While the method is theoretically amenable to on-policy training (via importance weighting or active sampling), all experiments are conducted in the fully offline setting. It would be valuable to demonstrate the effectiveness or limitations of on-policy extensions in sparse reward regimes such as docking or protein design.

**Quality:**

4

**Strengths And Weaknesses:**

**Strengths:**

1. **Clear motivation and well-scoped goal**: The paper addresses the problem of aligning molecular and protein generative models toward desirable properties — a key challenge in inverse design. The scope is clearly defined and practically relevant.

2. **Elegant and general algorithm (ERA)**: The proposed Energy Rank Alignment (ERA) framework is simple, principled, and broadly applicable. It sidesteps the complexities of reinforcement learning (e.g., PPO) and avoids the need for human-labeled preferences (as in DPO), while still offering strong theoretical guarantees. Its ability to decouple reward optimization (via β) and reference regularization (via γ) provides flexible control over the exploration–stability tradeoff.

3. **Fully offline training**: A major strength of ERA is its ability to work with fixed datasets and an explicit reward function, enabling completely offline training. This is especially useful in domains where querying the environment is expensive (e.g., docking or fitness evaluation). The method never samples from the evolving model during training, which distinguishes it clearly from on-policy methods.

4. **Strong performance across domains**: The framework is applied to four distinct settings — unprompted molecule generation, prompted molecule optimization, protein design, and docking — and consistently shows performance improvements over pretrained baselines. This demonstrates ERA’s robustness and cross-domain applicability.

5. **Theoretical clarity**: The paper offers a well-grounded derivation showing that ERA minimizes a KL divergence between the model’s predicted preference distribution and a softmax over rewards. This theoretical foundation lends credibility and interpretability to the method.

6. **Reasonable diversity and sample quality**: The model maintains chemical diversity and high reward without collapsing to a narrow output space. It performs comparably or better than other baselines such as REINVENT, MolRL-MGPT, and GFlowNet, with better sample efficiency in many cases.

---

**Weaknesses:**

1. **Validity tradeoff not analyzed**: Although validity is reported (and remains fairly high), it consistently drops after alignment — particularly in high-reward targets like LogP. The authors do not discuss the cause of this drop. A likely explanation is that ERA pushes the model toward high-reward but syntactically fragile or chemically unstable regions of the molecule space. This tradeoff between reward-seeking and validity deserves deeper analysis or mitigation strategies.

2. **No ablation on β and γ**: The method introduces two key hyperparameters — β for entropy shaping and γ for regularization toward the reference policy — but their effects are not thoroughly explored. A hyperparameter ablation would help guide users and support reproducibility.

3. **Random sampling strategy under-specified**: While ERA relies on sampling multiple outputs \( y, y' \) from the reference model for a fixed input \( x \), the paper does not clarify what sampling strategy is used (e.g., temperature, top-k, top-p). Since this randomness directly affects diversity and the quality of preference pairs, it would be helpful to include this detail or discuss its impact.

---

> ### Author Rebuttal · Authors · 2025-07-31
>
> Thank you for the thorough review and positive assessment of our work. We respond in detail below:
>
> > Sampling randomness during training: The ERA framework relies on sampling multiple outputs ( y, y' ) from the reference model ( \pi_{\text{ref}}(y|x) ) to construct preference pairs. However, the paper does not specify the sampling strategy (e.g., temperature, top-k, top-p) used during this process. Could the authors clarify what sampling method was used, whether temperature was tuned, and whether different sampling settings significantly affect performance, diversity, or training stability?
>
> We agree that this should be specified clearly in the paper. In order to ensure that our experiments focus on the hyperparameters of ERA and in all experiments we use the fairly standard practice of top-k sampling with $k=5$ and do not adjust the temperature $T=1.$
>
> > Validity drop after alignment: In several tasks, the validity of generated SMILES strings drops after applying ERA (e.g., LogP↑ task). Could the authors provide more insight into this phenomenon? Specifically, do invalid generations tend to cluster around certain reward-optimized motifs or SMILES patterns? Is there evidence that the model is being pushed toward syntactically fragile or chemically implausible regions of molecular space?
>
> Indeed, especially when the regularization $\gamma=0$, we see that ERA reduces validity of the generated SMILES. There are some limitations imposed by the fact that most of our downstream characterizations cannot be run on invalid SMILES strings because of the failure of RDkit to parse. However, in light of this interesting suggestion, we computed two measures of the diversity: an estimate of the Levenshtein distance and an estimate of the Shannon entropy for both valid and invalid molecules. These results are summarized in the table below:
>
> | Generations | Shannon Entropy | Levenshtein Distance |
> |-------------|------------------|----------------------|
> | Valid       | 2.2574           | 39.8225              |
> | Invalid     | 2.3177           | 56.1066              |
>
> The table above demonstrates properties of the valid and invalid generations from a model aligned using ERA for QED maximization for $\beta$=100.0, $\gamma$=0.1. Notably, the Shannon entropy of generated tokens and the mean Levenshtein distance between all of the generated SMILES is higher for the invalid outputs than the valid ones. These metrics suggest that there is in fact less clustering around one specific region of chemical space in these failure cases.
>
>
> > Analysis of failure cases: It would be helpful to understand the nature of invalid or low-reward generations. Could the authors provide examples or visualizations of failure cases (e.g., invalid SMILES, chemically unstable structures) and discuss what types of prompts or optimization targets tend to induce them?
>
> For the same QED maximization run ($\beta$=100.0, $\gamma$=0.1) as above, we found a significant failure mode to be that of missing tokens to close branches (67.6% of all invalid generations). Furthermore, failure to properly close rings also occurs in a signifcant amount of invalid generations (30.5%). One potential failure mode that seems to be relatively insignificant is that of failing to predict the stop token before our hard limit of 500 tokens, which occurs in fewer than 1% of invalid generations. Beyond this analysis, visual inspection of several failure cases does not appear to reveal any particular motif that is repeated among these examples beyond the model attempting to generate highly branched molecules or molecules with several rings.
>
>
> >Ablation of β and γ hyperparameters: The ERA algorithm introduces two key hyperparameters — β for entropy shaping and γ for reference policy regularization — yet there is limited analysis of their effects. Could the authors provide a more detailed ablation study or discussion showing how varying these parameters affects the tradeoffs between reward, validity, and diversity across tasks? Additionally, is there any task-specific guidance on selecting appropriate values?
>
> | $\beta$  | $\gamma$ | Validity (%) $\uparrow$ | Top-100 mean $\uparrow$ | Top-100 diversity $\downarrow$ |
> |---------|----------|--------------------------|--------------------------|-------------------------------|
> | 1       | 0        | 28.7                     | 0.598                    | 0.122                         |
> | 1       | 0.01     | 30.34                    | 0.611                    | 0.117                         |
> | 1       | 0.1      | 36.6                     | 0.685                    | 0.118                         |
> | 10      | 0        | 72.25                    | 0.930                    | 0.139                         |
> | 10      | 0.01     | 72.0                     | 0.932                    | 0.129                         |
> | 10      | 0.1      | 74.23                    | 0.933                    | 0.131                         |
> | 100     | 0        | 74.31                    | 0.934                    | 0.142                         |
> | 100     | 0.01     | 74.71                    | 0.934                    | 0.139                         |
> | 100     | 0.1      | 75.14                    | 0.935                    | 0.134                         |
>
>
> The table above contains descriptive statistics of 10,000 generated SMILES after aligning the molecular transformer model for QED maximization across several values of β and γ. For both β and γ, increasing its value led to an improvement in validity and performance in maximizing QED. These results demonstrate clear degradation in the model performance with certain choices of β (below 10), but these are not necessarily the case across all tasks. Table S4 in the appendix demonstrates how choices of β that were not appropriate for QED, like β=1.0, can be beneficial for other tasks such as MR and Ring Count. This study is being replicated for other optimization tasks discussed in the paper, and the results can be made available during the open discussion period at the reviewers' request.
>
> >Potential for constrained decoding: Given that validity tends to decline slightly under strong reward optimization, have the authors considered integrating grammar-based decoding, constrained decoding, or validity filters during generation? If so, what tradeoffs were observed? If not, could this be a promising direction to mitigate invalid generations while preserving ERA’s reward-guided behavior?
>
> This is an excellent suggestion, we have not implemented it because our models were still state-of-the-art in sample efficiency even accounting for the invalid generations, but it is certainly a good idea and one that we will investigate.

---

> > ### Comment · Reviewer_zXeq · 2025-08-06
> >
> > I appreciate the authors’ solid rebuttal addressing all reviewers’ comments. I believe they have taken a good step toward advancing domain‑specific applications and training strategies. Although the validity rate may not be the highest SOTA, I encourage the authors to also report the computational resources required for their proposed training strategy. That being said, I am happy to maintain my original score.

---

### Official Review · Reviewer_L7xi · 2025-07-03

**Clarity:** 2
**Significance:** 2
**Originality:** 2
**Rating:** 2
**Confidence:** 4

**Summary:**

The paper presents Energy Rank Alignment (ERA), a new algorithm for optimizing autoregressive models to efficiently explore chemical space and generate molecules with desired properties. Unlike RLHF, which depends on costly human preferences, or DPO, which uses implicit rewards, ERA employs an explicit reward function for gradient-based optimization. It aligns molecular generators to produce molecules with specific properties while ensuring diversity, converging to a Gibbs-Boltzmann distribution. Applied to small-molecule and protein sequence optimization, ERA demonstrates scalability and efficiency, advancing drug discovery and protein engineering.

**Questions:**

See weakness

**Ethical Concerns:**

["NO or VERY MINOR ethics concerns only"]

**Final Justification:**

My concerns have increased after reviewing the authors’ response. Respectfully, I am unsure whether the authors have a deep enough understanding of RL; some design choices may appear interesting but could actually harm performance instead or introduce unnecessary complexity. As they mention, “ERA works entirely offline”, which raises concerns about whether they fully understand the environment and how to take advantage of the given environment. It is also not fair to compare against DPO in a setting where an explicit reward function is available. A thorough understanding of each component in the PPO (and DPO) objective, as well as the potential empirical effects of altering them, is essential.

**Limitations:**

Yes

**Paper Formatting Concerns:**

Formatting is good.

**Quality:**

2

**Strengths And Weaknesses:**

Strengths:
1. Energy Rank Alignment (ERA) presents an innovative gradient-based optimization approach that uses an explicit reward function, eliminating the need for expensive reinforcement learning (e.g., RLHF) or implicit reward models (e.g., DPO).
2. The paper establishes a strong theoretical basis, demonstrating ERA’s convergence to a Gibbs-Boltzmann distribution and linking it to PPO and DPO.
3. ERA achieves higher sample efficiency than baselines like REINVENT and GraphiCA, effectively generating diverse molecules with desired properties.

Weaknesses:
1. Clarity Issue: Line 15’s claim that ERA “does not require reinforcement learning” is confusing, as introducing sample fluctuations around the maximal reward resembles RL techniques.
2. Unclear Need for Fluctuations: With an explicit reward function, the necessity of sampling fluctuations around the maximum reward is not well-justified.
3. Comparison to PPO: The paper does not clearly articulate ERA’s advantages over PPO when using an explicit reward function.
4. Missing Ablation Study: The paper lacks an ablation study comparing ERA, PPO, and DPO under explicit reward conditions, limiting insights into their relative performance.

---

> ### Author Rebuttal · Authors · 2025-07-31
>
> In response to the questions raised in your review, we explain below both experimental and theoretical characterization of ERA to clarify the relation to existing RL techniques.
>
> > “Clarity Issue: Line 15’s claim that ERA “does not require reinforcement learning” is confusing, as introducing sample fluctuations around the maximal reward resembles RL techniques.”
>
> We agree this has conceptual similarities to the entropic terms that are often used in reinforcement learning algorithms, like REINFORCE and PPO. The advantages of ERA include: it is a direct gradient-based procedure that works entirely offline, providing an important advantage for practical applications, and we do not need complicated protocols, like replay buffers or ad hoc objectives to promote sample diversity.
>
> >“With an explicit reward function, the necessity of sampling fluctuations around the maximum reward is not well-justified.”
>
> It is well-known that the reward functions used for molecular optimization are susceptible to reward hacking in which all the probability associated with a given task concentrates on a single motif. Just as in classical RL approaches, adding an ad hoc entropy term can mitigate this undesired behavior. **The theoretical justification for adding an entropy term is to increase sample diversity in a controllable fashion, ensuring near optimality, but avoiding greedy optimization.** When the number of observed comparisons is small, DPO maximizes the margin between pairs of samples. In our experiments, this leads to both low validity of the generated SMILES and low diversity because the probability concentrates on the “winning” molecules and diversity collapses:
> When the preference dataset only has one observation $\boldsymbol{y}\succ \boldsymbol{y}'$ per prompt $\boldsymbol{x}$, the optimal policy requires that
> \begin{equation}
>     \pi_{\star}^{\textrm{DPO}}(\boldsymbol{y}|\boldsymbol{x}) = 1 \quad \textrm{and} \quad \pi_{\star}^{\textrm{DPO}}(\boldsymbol{y}'|\boldsymbol{x}) = 0.
> \end{equation}
> The sampled gradients of the objective used for DPO are proportional to the implicit reward discrepancy,
> \begin{equation}
>     \nabla_{\boldsymbol{\theta}} \hat{\mathcal{L}}^{\textrm{DPO}}(\boldsymbol{y}, \boldsymbol{y}', \boldsymbol{x}) = \sigma \left( \beta^{-1}\gamma \left[\log \frac{\pi(\boldsymbol{y}'|\boldsymbol{x})}{\pi_{\rm ref}(\boldsymbol{y}'|\boldsymbol{x})} - \log \frac{\pi(\boldsymbol{y}|\boldsymbol{x})}{\pi_{\rm ref}(\boldsymbol{y}|\boldsymbol{x})} \right] \right) \nabla_{\boldsymbol{\theta}} \log \frac{\pi(\boldsymbol{y}|\boldsymbol{x})}{\pi(\boldsymbol{y}'|\boldsymbol{x})},
> \end{equation}
> which when $\pi(\boldsymbol{y}'|\boldsymbol{x}) \to 0$, could lead to instability as $-\log \pi(\boldsymbol{y}'|\boldsymbol{x}) \to \infty.$
> On the other hand, the ERA gradients are scaled by the relative preference discrepancy. The advantage of a reward model becomes apparent because the optimum of $\mathcal{L}^{\textrm{ERA}}$ will not lead to policies in which $\textrm{supp}(\pi)$ degrades unless the energy becomes infinite. This could alternatively be implemented by taking $\beta\to\infty$. Choosing an appropriate reward model, hence, gives the flexibility to control instability if it becomes problematic.
>
> > “Comparison to PPO: The paper does not clearly articulate ERA’s advantages over PPO when using an explicit reward function.”
>
> The purpose of ERA is to optimize with a molecular reward function while preserving sample diversity. We compare ERA to several reinforcement learning strategies throughout, which optimize objectives that are commonly used in PPO.
>
>  - REINVENT uses a policy gradient algorithm on the PPO objective described below. Our benchmarks consistently outperform REINVENT.
>  - MolRL-MGPT uses multi-agent reinforcement learning with an objective identical to the one used in REINVENT. The objective requires manually balancing several terms to mitigate overfitting. **The results of these comparisons can be seen in Table 1 of our manuscript.**
>
> This objective we introduce differs from the regularized reward loss conventionally used for PPO: the minimizer of the PPO objective is also a Gibbs-Boltzmann measure, explicitly,
> \begin{equation}
>     \pi_{\star}^{(\textrm{PPO})} \propto \exp \left[ -\frac{\beta}{\gamma} U(\boldsymbol{x},\boldsymbol{y}) + \log \pi_{\rm ref}(\boldsymbol{y}|\boldsymbol{x}) \right].
> \end{equation}
> Here, the KL-regularization corresponds to an energy shift, as in our objective, but there is no limit in which the ideal distribution $\pi\propto e^{-\beta U}$ is obtained for the PPO objective.
> This is in stark contrast to our approach, which recovers the ideal distribution as $\gamma\to0.$
> Furthermore, while our approach allows for a direct gradient-based optimization using the direct estimator introduced in the paper, PPO is implemented using an actor-critic framework that is difficult to tune [cf. e.g., rafailov_direct_2023, casper_open_2023].
>
> > “The paper lacks an ablation study comparing ERA, PPO, and DPO under explicit reward conditions, limiting insights into their relative performance.”
>
> We provide here an ablation study of the $\beta$ and $\gamma$ hyperparameters involved in ERA:
>
> | $\beta$  | $\gamma$ | Validity (%) $\uparrow$ | Top-100 mean $\uparrow$ | Top-100 diversity $\downarrow$ |
> |---------|----------|--------------------------|--------------------------|-------------------------------|
> | 1       | 0        | 28.7                     | 0.598                    | 0.122                         |
> | 1       | 0.01     | 30.34                    | 0.611                    | 0.117                         |
> | 1       | 0.1      | 36.6                     | 0.685                    | 0.118                         |
> | 10      | 0        | 72.25                    | 0.930                    | 0.139                         |
> | 10      | 0.01     | 72.0                     | 0.932                    | 0.129                         |
> | 10      | 0.1      | 74.23                    | 0.933                    | 0.131                         |
> | 100     | 0        | 74.31                    | 0.934                    | 0.142                         |
> | 100     | 0.01     | 74.71                    | 0.934                    | 0.139                         |
> | 100     | 0.1      | 75.14                    | 0.935                    | 0.134                         |
>
> The table above contains descriptive statistics of 10,000 generated SMILES after aligning the molecular transformer model for QED maximization across several values of β and γ. For both β and γ, increasing its value led to an improvement in validity and performance in maximizing QED. These results demonstrate clear degradation in the model performance with certain choices of β (below 10), but these are not necessarily the case across all tasks. Table S4 demonstrates how choices of β that were not appropriate for QED, like β=1.0, can be beneficial for other tasks such as MR and Ring Count. This study is being replicated for other optimization tasks, and the results can be made available during the open discussion period.
>
> For DPO, we note that the DPO objective does not contain an explicit reward and instead only maximizes the margins between preferred and dis-preferred samples. Additional care needs to be taken when comparing DPO and ERA hyperparameters because $\beta_{DPO} \neq \beta_{ERA}$ due to a difference in definition. Specifically, the minimizer of the DPO objective is
> $$
> \pi_{DPO}(x, y) \propto \exp(-\beta_{DPO}^{-1}r(x, y) + \log \pi_{ref})
> $$
> Whereas for ERA, the minimizer is
> $$
> \pi_{ERA}(x, y) \propto \exp\left(\frac{-\beta_{ERA}}{1 + \gamma}r(x, y) + \frac{\gamma}{1 + \gamma}\log \pi_{ref}\right)
> $$
> The formulation of the DPO objective does not permit explicitly scaling the contribution of the regularization term against the prior while ERA permits this. This also implies that
> $$
> \beta_{ERA} = \frac{1 + \gamma}{\beta_{DPO}}
> $$
> We take $\gamma$ to be relatively large since as $\gamma \to \infty$, $\frac{\gamma}{1+\gamma} \to 1$. In our comparisons, we choose $\gamma = 9$. The tables below are arranged such that the corresponding rows can be compared directly against each other. We did not scan any $\beta < 0.1$ since the validity in those cases was low for both methods:
>
> **DPO results**
> | $\beta_{DPO}$ | Validity (%) $\uparrow$ | Top-100 mean $\uparrow$ | Top-100 diversity $\downarrow$ |
> | -------- | -------- | -------- | -------- |
> | 100      | 86.91     | 0.936     | 0.131
> | 10       | 80.92     | 0.943     | 0.134
> | 1        | 82.23     | 0.943     | 0.156
> | 0.1      | 54.46     | 0.947     | 0.174
>
> **ERA results ($\gamma = 9$)**
> | $\beta_{ERA}$ | Validity (%) $\uparrow$ | Top-100 mean $\uparrow$ | Top-100 diversity $\downarrow$ |
> |---------------|--------------------------|--------------------------|---------------------------------|
> | 0.1           | 84.78                    | 0.939                    | 0.149                           |
> | 1             | 85.73                    | 0.936                    | 0.145                           |
> | 10            | 79.07                    | 0.932                    | 0.141                           |
> | 100           | 88.83                    | 0.941                    | 0.145                           |
>
> The tables above contain descriptive statistics of 10,000 generated SMILES after aligning the molecular transformer model for QED maximization using either DPO or ERA across several matched values of $\beta$. Due to parameter matching, we expect ERA and DPO to be similar here: validity is comparable, with ERA surpassing DPO at higher $\beta_{ERA}$, as well as the top-100 average QED. **ERA consistently generates diverse molecules across values of $\beta_{ERA}$ as opposed to DPO**, cf. lower similarity among the top-100 generations for $\beta_{ERA} = 10$ and $\beta_{ERA} = 100$ because ERA objective promotes diverse generations, even without a strong prior.

---

> ### Comment · Reviewer_L7xi · 2025-08-09
>
> Thank you for your response. I have reviewed it carefully and revisited your paper, but my main concern remains unaddressed — and has, in fact, been heightened.
>
> 1. **"...the advantages of ERA include: it is a direct gradient-based procedure that works entirely offline, providing an important advantage for practical applications..."**
> If you have an explicit reward function with a low interaction cost, why design an algorithm that operates entirely offline and consider that an advantage? This approach does not fully leverage the explicit reward function or the environmental conditions to improve performance. Theoretically, given the environment, it will be hard to achieve optimal performance.
>
> 2.  **"compare with DPO and PPO"**
> DPO lacks an explicit reward function and is well-suited for preference-based reward scenarios. In your environment, however, an explicit reward function is available, yet your approach is kind of reducing this signal to preference learning. Theoretically, with appropriate hyperparameter tuning, PPO would likely outperform your algorithm in this setting.
>
> 3.  **"compared with REINVENT"**
> The REINVENT paper employs a specialized pair-based pretraining approach that differs from yours, so this does not constitute a convincing ablation study. Outperforming REINVENT does not necessarily imply that ERA outperforms PPO in your setting.
>
> 4. **"compared with PPO objective"**
> The regularization term in the PPO objective is intended to prevent performance instability (e.g., sudden drops). Slightly modifying this term does not necessarily make your objective substantially different or better. Setting $\gamma \rightarrow 0$ could potentially lead to performance degradation.

---

### Note · Authors · 2025-08-13

We thank the reviewers for their continued engagement. We were unable to address a number of very last-minute comments, and now do so below. We believe we have fully addressed the stated concerns of the reviewers, including direct comparison with PPO. Our experiments show that ERA outperforms existing methods for molecular optimization and does so with exceptional computational efficiency. We maintain that ERA possesses distinctive theoretical and practical advantages over competing methods and our experiments convincingly substantiate this claim.

 1. We continue to assert that ERA is more principled than DPO or RL for tasks where a reward is directly measured (e.g., experiments or expensive computational tasks, like our directed evolution benchmark). The offline nature of the algorithm is advantageous in scenarios where the reward is not necessarily cheap to evaluate. In fact, we show that ERA demonstrates exceptional computational efficiency both in terms of the compute required for inference on a single GPU and the sample efficiency as measured by the AUC of the top-10 statistics against the number of oracle calls (see the response to reviewer EGLr).

 2. We compared extensively to reward-based RL algorithms, inlcuding REINVENT and MolMolRL-mGPT, as well as actor-critic type methods like GFlowNet in our paper. ERA already outperforms these techniques, which also leverage the explicit reward.

 3. Evidence from the literature generally shows that PPO implementations underperform REINVENT. We see that ERA outperforms PPO on benchmark tasks we considered, cf. [https://pubs.acs.org/doi/pdf/10.1021/acs.jcim.4c00895]. We significantly outperform both PPO and PPOD, a modified version of PPO for sparse reward settings, on GSK3$\beta$ and JNK3 inhibitor optimization tasks, as seen in the table of the top-10 AUC metric below:


| Method | GSK3$\beta$ $\uparrow$ | JNK3 $\uparrow$ |
| -------- | -------- | -------- |
| **ERA**     | **0.985**      | **0.989**    |
| PPO     | 0.90     |      0.80|
| PPOD     | 0.92     | 0.87     |


 4. The distinction with PPO is not only the objective: these are indeed similar. While our regularization is more easily and independently tuned than in PPO-type objectives, our optimization pipeline is completely different leveraging a simple, gradient-based loss than can be optimized in experimental workflows (e.g., 96 well plates of that produce a fixed set of experimental observations, as in our directed evolution benchmarks)

---

### Decision · Program_Chairs · 2025-09-17

**Decision:**

Accept (poster)

**Comment:**

This paper introduces energy rank alignment, an approach for to use explicit reward function to produce a gradient-based objective for auto-regressive policies; the authors show how this can be used in transformer-based generative models in molecular applications. The paper had extensive experimentation and evaluation in the initial submission. The most critical issues the reviewers raised concerned the comparisons to several baselines, ablations and presentation, all of which were thoroughly addressed during. the discussion phase. The paper presents an interesting new method, which may have impact beyond molecular machine learning applications in the future.